# A Refutation of Shapley Values for Explainability

## Abstract

Recent work demonstrated the existence of Boolean functions for which Shapley values provide misleading information about the relative importance of features in rule-based explanations. Such misleading information was broadly categorized into a number of possible issues. Each of those issues relates with features being relevant or irrelevant for a prediction, and all are significant regarding the inadequacy of Shapley values for rule-based explainability. This earlier work devised a brute-force approach to identify Boolean functions, defined on small numbers of features, and also associated instances, which displayed such inadequacy-revealing issues, and so served as evidence to the inadequacy of Shapley values for rule-based explainability. However, an outstanding question is how frequently such inadequacy-revealing issues can occur for Boolean functions with arbitrary large numbers of features. It is plain that a brute-force approach would be unlikely to provide insights on how to tackle this question. This paper answers the above question by proving that, for any number of features, there exist Boolean functions that exhibit one or more inadequacy-revealing issues, thereby contributing decisive arguments against the use of Shapley values as the theoretical underpinning of feature-attribution methods in explainability.

## 1 Introduction

Feature attribution is one of the most widely used approaches in machine learning (ML) explainability, begin implemented with a variety of different methods [64, 56, 57]. Moreover, the use of Shapley values [60] for feature attribution ranks among the most popular solutions [64, 65, 48, 17, 47], offering a widely accepted theoretical justification on how to assign importance to features in machine learning (ML) model predictions. Despite the success of using Shapley values for explainability, it is also the case that their exact computation is in general intractable [8, 21, 22], with tractability results for some families of boolean circuits [8]. As a result, a detailed assessment of the rigor of feature attribution methods based on Shapley values, when compared with exactly computed Shapley values has not been investigated. Furthermore, the definition Shapley values (as well as its use in explainability) is purely axiomatic, i.e. there exists *no* formal proof that Shapley values capture any specific properties related with explainability (even if defining such properties might prove elusive).

Feature selection represents a different alternative to feature attribution. The goal of feature selection is to select a set of features as representing the reason for a prediction, i.e. if the selected features take their assigned values, then the prediction cannot be changed. There are rigorous and non-rigorous approaches for selecting the features that explain a prediction. This paper considers rigorous (or model-precise) approaches for selecting such features. Furthermore, it should be plain that feature selection must aim for irredundancy, since otherwise it would suffice to report all features as the explanation. Given the universe of possible irreducible sets of feature selections that explain a prediction, the features that do not occur in *any* such set are deemed *irrelevant* for a prediction; otherwise features that occur in one or more feature selections are deemed *relevant*.

Since both feature attribution and feature selection measure contributions of features to explanations, one would expect that the two approaches were related. However, this is not the case. Recent

Submitted to 37th Conference on Neural Information Processing Systems (NeurIPS 2023). Do not distribute.

work [35] observed that feature attribution based on Shapley values could produce *misleading information* about features, in that irrelevant features (for feature selection) could be deemed more important (in terms of feature attribution) than relevant features (also for feature selection). Clearly, misleading information about the relative importance of features can easily induce human decision makers in error, by suggesting the *wrong* features as those to analyze in greater detail. Furthermore, situations where human decision makers can be misled are inadmissible in high-risk or safety-critical uses of ML. Furthermore, a number of possible misleading issues of Shapley values for explainability were identified [35], and empirically demonstrated to occur for some boolean functions. The existence in practice of those misleading issues with Shapley values for explainability is evidently problematic for their use as the theoretical underpinning of feature attribution methods.

However, earlier work [35] used a brute-force method to identify boolean functions, defined on a very small number of variables, where the misleading issues could be observed. A limitation of this earlier work [35] is that it offered no insights on how general the issues with Shapley values for explainability are. For example, it could be the case that the identified misleading issues might only occur for functions defined on a very small number of variables, or in a negligible number of functions, among the universe of functions defined on a given number of variables. If that were to be the case, then the issues with Shapley values for explainability might not be that problematic.

This paper proves that the identified misleading issues with Shapley values for explainability are much more general that what was reported in earlier work [35]. Concretely, the paper proves that, for any number of features larger than a small $k$ (either 2 or 3), one can easily construct functions which exhibit the identified misleading issues. The main implication of our results is clear: *the use of Shapley values for explainability can, for an arbitrary large number of boolean (classification) functions, produce misleading information about the relative importance of features.*

**Organization.** The paper is organized as follows. Section 2 introduces the notation and definitions used throughout the paper. Section 3 revisits and extends the issues with Shapley values for explainability reported in earlier work [35], and illustrates the existence of those issues in a number of motivating example boolean functions. Section 4 presents the paper's main results, proving that all the issues with Shapley values for explainability reported in earlier work [35] occur for boolean functions with arbitrarily larger number of variables. (Due to lack of space, the detailed proofs are all included in Appendix A, and the paper includes only brief insights into those proofs.) Also, the proposed constructions offer ample confidence that the number of functions displaying one or more of the issues is significant. Section 5 concludes the paper.

## 2   Preliminaries

**Boolean functions.** Let $\mathbb{B} = \{0, 1\}$. The results in the paper consider boolean functions, defined on $m$ boolean variables, i.e. $\kappa : \mathbb{B}^m \to \mathbb{B}$. (The fact that we consider only boolean functions does not restrict in the significance of the results.)

In the rest of the paper, we will use the boolean functions shown in Figure 1, which are represented by truth tables. The highlighted rows will serve as concrete examples throughout.

**Classification in ML.** A classification problem is defined on a set of features $\mathcal{F} = \{1, \ldots, m\}$, each with domain $\mathbb{D}_i$, and a set of classes $\mathcal{K} = \{c_1, c_2, \ldots, c_K\}$. (As noted above, we will assume $\mathbb{D}_i = \mathbb{B}$ for $1 \leq i \leq m$, but domains could be categorical or ordinal. Also, we will assume $\mathcal{K} = \mathbb{B}$.) Feature space $\mathbb{F}$ is defined as the cartesian product of the domains of the features, in order: $\mathbb{F} = \mathbb{D}_1 \times \cdots \times \mathbb{D}_m$, which will be $\mathbb{B}^m$ throughout the paper. A classification function is a non-constant map from feature space into the set of classes, $\kappa : \mathbb{F} \to \mathcal{K}$. (Clearly, a classifier would be useless if the classification function were constant.) Throughout the paper, we will not distinguish between classifiers and boolean functions. An instance is a pair $(\mathbf{v}, c)$ representing a point $\mathbf{v} = (v_1, \ldots, v_m)$ in feature space, and the classifier's prediction, i.e. $\kappa(\mathbf{v}) = c$. Moreover, we let $\mathbf{x} = (x_1, \ldots, x_m)$ denote an arbitrary point in the feature space. Abusing notation, we will also use $\mathbf{x}_{a..b}$ to denote $x_a, \ldots, x_b$, and $\mathbf{v}_{a..b}$ to denote $v_a, \ldots, v_b$. Finally, a classifier $\mathcal{M}$ is a tuple $(\mathcal{F}, \mathbb{F}, \mathcal{K}, \kappa)$. In addition, an explanation problem $\mathcal{E}$ is a tuple $(\mathcal{M}, (\mathbf{v}, c))$, where $\mathcal{M} = (\mathcal{F}, \mathbb{F}, \mathcal{K}, \kappa)$ is a classifier.

**Shapley values for explainability.** Shapley values were first introduced by L. Shapley [60] in the context of game theory. Shapley values have been extensively used for explaining the predictions of ML models, e.g. [64, 65, 20, 48, 15, 52, 62, 69], among a vast number of recent examples. The complexity of computing Shapley values (as proposed in SHAP [48]) has been studied in recent

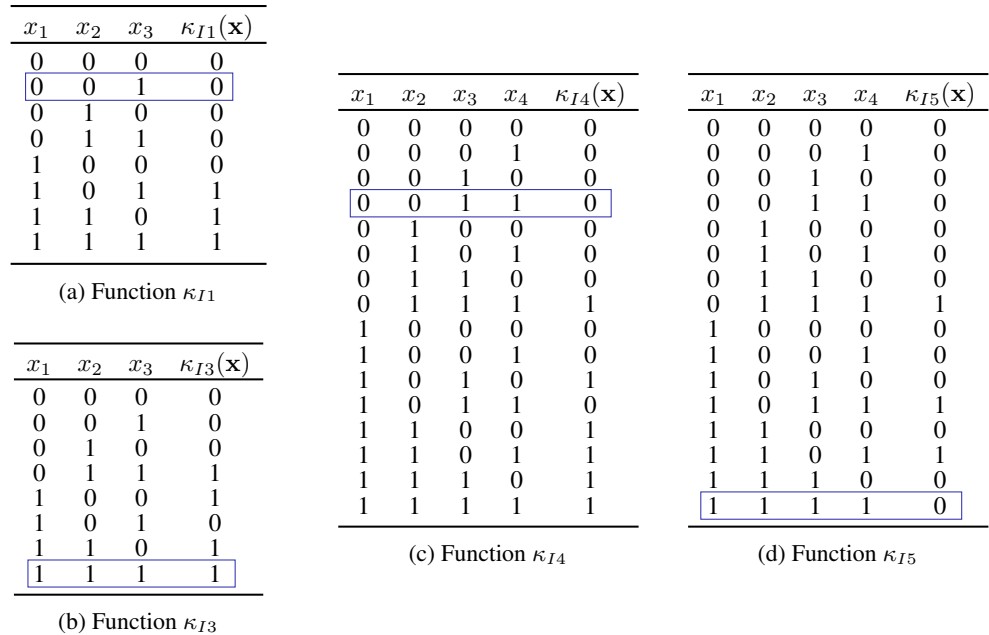

| $x_1$ | $x_2$ | $x_3$ | $\kappa_{I1}(\mathbf{x})$ |
|---|---|---|---|
| 0 | 0 | 0 | 0 |
| 0 | 0 | 1 | 0 |
| 0 | 1 | 0 | 0 |
| 0 | 1 | 1 | 0 |
| 1 | 0 | 0 | 0 |
| 1 | 0 | 1 | 1 |
| 1 | 1 | 0 | 1 |
| 1 | 1 | 1 | 1 |

(a) Function $\kappa_{I1}$

| $x_1$ | $x_2$ | $x_3$ | $\kappa_{I3}(\mathbf{x})$ |
|---|---|---|---|
| 0 | 0 | 0 | 0 |
| 0 | 0 | 1 | 0 |
| 0 | 1 | 0 | 0 |
| 0 | 1 | 1 | 1 |
| 1 | 0 | 0 | 1 |
| 1 | 0 | 1 | 0 |
| 1 | 1 | 0 | 1 |
| 1 | 1 | 1 | 1 |

(b) Function $\kappa_{I3}$

| $x_1$ | $x_2$ | $x_3$ | $x_4$ | $\kappa_{I4}(\mathbf{x})$ |
|---|---|---|---|---|
| 0 | 0 | 0 | 0 | 0 |
| 0 | 0 | 0 | 1 | 0 |
| 0 | 0 | 1 | 0 | 0 |
| 0 | 0 | 1 | 1 | 0 |
| 0 | 1 | 0 | 0 | 0 |
| 0 | 1 | 0 | 1 | 0 |
| 0 | 1 | 1 | 0 | 0 |
| 0 | 1 | 1 | 1 | 1 |
| 1 | 0 | 0 | 0 | 0 |
| 1 | 0 | 0 | 1 | 0 |
| 1 | 0 | 1 | 0 | 1 |
| 1 | 0 | 1 | 1 | 0 |
| 1 | 1 | 0 | 0 | 1 |
| 1 | 1 | 0 | 1 | 1 |
| 1 | 1 | 1 | 0 | 1 |
| 1 | 1 | 1 | 1 | 1 |

(c) Function $\kappa_{I4}$

| $x_1$ | $x_2$ | $x_3$ | $x_4$ | $\kappa_{I5}(\mathbf{x})$ |
|---|---|---|---|---|
| 0 | 0 | 0 | 0 | 0 |
| 0 | 0 | 0 | 1 | 0 |
| 0 | 0 | 1 | 0 | 0 |
| 0 | 0 | 1 | 1 | 0 |
| 0 | 1 | 0 | 0 | 0 |
| 0 | 1 | 0 | 1 | 0 |
| 0 | 1 | 1 | 0 | 0 |
| 0 | 1 | 1 | 1 | 1 |
| 1 | 0 | 0 | 0 | 0 |
| 1 | 0 | 0 | 1 | 0 |
| 1 | 0 | 1 | 0 | 0 |
| 1 | 0 | 1 | 1 | 1 |
| 1 | 1 | 0 | 0 | 0 |
| 1 | 1 | 0 | 1 | 1 |
| 1 | 1 | 1 | 0 | 0 |
| 1 | 1 | 1 | 1 | 0 |

(d) Function $\kappa_{I5}$

Figure 1: Example functions for issues I1, I3, I4, I5, respectively $\kappa_{I1}$, $\kappa_{I3}$, $\kappa_{I4}$, $\kappa_{I5}$

years [8, 21, 7, 22]. This section provides a brief overview of Shapley values. Throughout the section, we adapt the notation used in recent work [8, 7], which builds on the work of [48].

Let $\Upsilon : 2^{\mathcal{F}} \to 2^{\mathbb{F}}$ be defined by[1],

$$\Upsilon(\mathcal{S}; \mathbf{v}) = \{\mathbf{x} \in \mathbb{F} \mid \wedge_{i \in \mathcal{S}} x_i = v_i\} \tag{1}$$

i.e. for a given set $\mathcal{S}$ of features, and parameterized by the point $\mathbf{v}$ in feature space, $\Upsilon(\mathcal{S}; \mathbf{v})$ denotes all the points in feature space that have in common with $\mathbf{v}$ the values of the features specified by $\mathcal{S}$.

Also, let $\phi : 2^{\mathcal{F}} \to \mathbb{R}$ be defined by,

$$\phi(\mathcal{S}; \mathcal{M}, \mathbf{v}) = \frac{1}{2^{|\mathcal{F} \setminus \mathcal{S}|}} \sum_{\mathbf{x} \in \Upsilon(\mathcal{S}; \mathbf{v})} \kappa(\mathbf{x}) \tag{2}$$

For the purposes of this paper, we consider solely a uniform input distribution, and so the dependency on the input distribution is not accounted for. A more general formulation is considered in related work [8, 7]. However, assuming a uniform distribution suffices for the purposes of this paper. As a result, given a set $\mathcal{S}$ of features, $\phi(\mathcal{S}; \mathcal{M}, \mathbf{v})$ represents the average value of the classifier over the points of feature space represented by $\Upsilon(\mathcal{S}; \mathbf{v})$.

Finally, let $\mathsf{Sv} : \mathcal{F} \to \mathbb{R}$ be defined by[2],

$$\mathsf{Sv}(i; \mathcal{M}, \mathbf{v}) = \sum_{\mathcal{S} \subseteq (\mathcal{F} \setminus \{i\})} \frac{|\mathcal{S}|!(|\mathcal{F}| - |\mathcal{S}| - 1)!}{|\mathcal{F}|!} (\phi(\mathcal{S} \cup \{i\}; \mathcal{M}, \mathbf{v}) - \phi(\mathcal{S}; \mathcal{M}, \mathbf{v})) \tag{3}$$

Given an instance $(\mathbf{v}, c)$, the Shapley value assigned to each feature measures the *contribution* of that feature with respect to the prediction. A positive/negative value indicates that the feature can contribute to changing the prediction, whereas a value of 0 indicates no contribution.

**Example 1.** We consider the example boolean functions of Figure 1. If the functions are represented by a truth table, then the Shapley values can be computed in polynomial time on the size of the

---

[1] When defining concepts, we will show the necessary parameterizations. However, in later uses, those parameterizations will be omitted, for simplicity.

[2] We distinguish $\mathsf{SHAP}(\cdot; \cdot, \cdot)$ from $\mathsf{Sv}(\cdot; \cdot, \cdot)$. Whereas $\mathsf{SHAP}(\cdot; \cdot, \cdot)$ represents the value computed by the tool SHAP [48], $\mathsf{Sv}(\cdot; \cdot, \cdot)$ represents the Shapley value in the context of (feature attribution based) explainability, as studied in a number of works [64, 65, 48, 8, 21, 22]. Thus, $\mathsf{SHAP}(\cdot; \cdot, \cdot)$ is a heuristic approximation of $\mathsf{Sv}(\cdot; \cdot, \cdot)$.

truth table, since the number of subsets considered in (3) is also polynomial on the size of the truth table [35]. (Observe that for each subset used in (3), we can use the truth table for computing the average values in (2).) For example, for $\kappa_{I1}$ and for the point in feature space $(0, 0, 1)$, one can compute the following Shapley values: $\mathsf{Sv}(1) = -0.417$, $\mathsf{Sv}(2) = -0.042$, and $\mathsf{Sv}(3) = 0.083$.

**Logic-based explanations.** There has been recent work on developing formal definitions of explanations. One type of explanations are *abductive explanations* [37] (AXp), which corresponds to a PI-explanations [61] in the case of boolean classifiers. AXp's represent prime implicants of the discrete-valued classifier function (which computes the predicted class). AXp's can also be viewed as an instantiation of logic-based abduction [24, 59, 13, 23]. Throughout this paper we will opt to use the acronym AXp to refer to abductive explanations.

Let us consider a given classifier, computing a classification function $\kappa$ on feature space $\mathbb{F}$, a point $\mathbf{v} \in \mathbb{F}$, with prediction $c = \kappa(\mathbf{v})$, and let $\mathcal{X}$ denote a subset of the set of features $\mathcal{F}$, $\mathcal{X} \subseteq \mathcal{F}$. $\mathcal{X}$ is a weak AXp for the instance $(\mathbf{v}, c)$ if,

$$\mathsf{WAXp}(\mathcal{X}; \mathcal{M}, \mathbf{v}) \quad := \quad \forall (\mathbf{x} \in \mathbb{F}). \left[ \bigwedge_{i \in \mathcal{X}} (x_i = v_i) \right] \rightarrow (\kappa(\mathbf{x}) = c) \tag{4}$$

where $c = \kappa(\mathbf{v})$. Thus, given an instance $(\mathbf{v}, c)$, a (weak) AXp is a subset of features which, if fixed to the values dictated by $\mathbf{v}$, then the prediction is guaranteed to be $c$, independently of the values assigned to the other features.

Moreover, $\mathcal{X} \subseteq \mathcal{F}$ is an AXp if, besides being a weak AXp, it is also subset-minimal, i.e.

$$\mathsf{AXp}(\mathcal{X}; \mathcal{M}, \mathbf{v}) \quad := \quad \mathsf{WAXp}(\mathcal{X}; \mathcal{M}, \mathbf{v}) \wedge \forall (\mathcal{X}' \subsetneq \mathcal{X}). \neg \mathsf{WAXp}(\mathcal{X}'; \mathcal{M}, \mathbf{v}) \tag{5}$$

Observe that an AXp can be viewed as a possible irreducible answer to a "**Why?**" question, i.e. why is the classifier's prediction $c$? It should be plain in this work, but also in earlier work, that the representation of AXp's using subsets of features aims at simplicity. The sufficient condition for the prediction is evidently the conjunction of literals associated with the features contained in the AXp.

**Example 2.** Similar to the computation of Shapley values, given a truth table representation of a function, and for a given instance, there is a polynomial-time algorithm for computing the AXp's [35]. For example, for function $\kappa_{I4}$ (see Figure 1c), and for the instance $((0, 0, 1, 1), 0)$, it can be observed that, if features 3 and 4 are allowed to take other values, the prediction remains at 0. Hence, $\{1, 2\}$ is an WAXp, which is easy to conclude that it is also an AXp. When interpreted as a rule, the AXp would yield the rule:

$$\text{IF} \quad \neg x_1 \wedge \neg x_2 \quad \text{THEN} \quad \kappa(\mathbf{x}) = 0$$

In a similar way, if features 1 and 3 are allowed to take other values, the prediction remains at 0. Hence, $\{2, 4\}$ is another WAXp (which can easily be shown to be an AXp). Furthermore, considering all other possible subsets of fixed features, allows us to conclude that there are no more AXp's.

Similarly to the case of AXp's, one can define (weak) contrastive explanations (CXp's) [53, 36]. $\mathcal{Y} \subseteq \mathcal{F}$ is a weak CXp for the instance $(\mathbf{v}, c)$ if,

$$\mathsf{WCXp}(\mathcal{Y}; \mathcal{M}, \mathbf{v}) \quad := \quad \exists (\mathbf{x} \in \mathbb{F}). \left[ \bigwedge_{i \notin \mathcal{Y}} (x_i = v_i) \right] \wedge (\kappa(\mathbf{x}) \neq c) \tag{6}$$

(As before, for simplicity we keep the parameterization of $\mathsf{WCXp}$ on $\kappa$, $\mathbf{v}$ and $c$ implicit.) Thus, given an instance $(\mathbf{v}, c)$, a (weak) CXp is a subset of features which, if allowed to take any value from their domain, then there is an assignment to the features that changes the prediction to a class other than $c$, this while the features not in the explanation are kept to their values.

Furthermore, a set $\mathcal{Y} \subseteq \mathcal{F}$ is a CXp if, besides being a weak CXp, it is also subset-minimal, i.e.

$$\mathsf{CXp}(\mathcal{Y}; \mathcal{M}, \mathbf{v}) \quad := \quad \mathsf{WCXp}(\mathcal{Y}; \mathcal{M}, \mathbf{v}) \wedge \forall (\mathcal{Y}' \subsetneq \mathcal{Y}). \neg \mathsf{WCXp}(\mathcal{Y}'; \mathcal{M}, \mathbf{v}) \tag{7}$$

A CXp can be viewed as a possible irreducible answer to a "**Why Not?**" question, i.e. why isn't the classifier's prediction a class other than $c$?

**Example 3.** For the example function $\kappa_{I4}$ (see Figure 1c), and instance $((0, 0, 1, 1), 0)$, if we fix features 1, 3 and 4, respectively to 0, 1 1, then by allowing feature 2 to change value, we see that the prediction changes, e.g. by considering the point $(0, 1, 1, 1)$ with prediction 1. Thus, $\{2\}$ is a CXp. In a similar way, by fixing the features 2 and 3, respectively to 0 and 1, then by allowing features 1 and 4 to change value, we conclude that the prediction changes. Hence, $\{1, 4\}$ is also a CXp.

The sets of AXp's and CXp's are defined as follows:

$$\begin{aligned} \mathbb{A}(\mathcal{E}) &= \{ \mathcal{X} \subseteq \mathcal{F} \,|\, \mathsf{AXp}(\mathcal{X}; \mathcal{M}, \mathbf{v}) \} \\ \mathbb{C}(\mathcal{E}) &= \{ \mathcal{Y} \subseteq \mathcal{F} \,|\, \mathsf{CXp}(\mathcal{Y}; \mathcal{M}, \mathbf{v}) \} \end{aligned} \tag{8}$$

(The parameterization on $\mathcal{M}$ and $\mathbf{v}$ is unnecessary, since the explanation problem $\mathcal{E}$ already accounts for those.) Moreover, let $F_{\mathbb{A}}(\mathcal{E}) = \cup_{\mathcal{X} \in \mathbb{A}(\mathcal{E})}\mathcal{X}$ and $F_{\mathbb{C}}(\mathcal{E}) = \cup_{\mathcal{Y} \in \mathbb{C}(\mathcal{E})}\mathcal{Y}$. $F_{\mathbb{A}}(\mathcal{E})$ aggregates the features occurring in any abductive explanation, whereas $F_{\mathbb{C}}(\mathcal{E})$ aggregates the features occurring in any contrastive explanation. In addition, minimal hitting set duality between AXp's and CXp's [36] yields the following result[3].

**Proposition 1.** $F_{\mathbb{A}}(\mathcal{E}) = F_{\mathbb{C}}(\mathcal{E})$.

**Feature (ir)relevancy in explainability.** Given the definitions above, we have the following characterization of features [33, 34, 32]:

1. A feature $i \in \mathcal{F}$ is *necessary* if $\forall(\mathcal{X} \in \mathbb{A}(\mathcal{E})).i \in \mathcal{X}$.
2. A feature $i \in \mathcal{F}$ is *relevant* if $\exists(\mathcal{X} \in \mathbb{A}(\mathcal{E})).i \in \mathcal{X}$.
3. A feature is *irrelevant* if it is not relevant, i.e. $\forall(\mathcal{X} \in \mathbb{A}(\mathcal{E})).i \notin \mathcal{X}$.

By Proposition 1, the definitions of necessary and relevant feature could instead use $\mathbb{C}(\mathcal{E})$. Throughout the paper, we will use the predicate Irrelevant($i$) which holds true if feature $i$ is irrelevant, and predicate Relevant($i$) which holds true if feature $i$ is relevant. Furthermore, it should be noted that feature irrelevancy is a fairly demanding condition in that, a feature $i$ is irrelevant if it is not included in *any* subset-minimal set of features that is sufficient for the prediction.

**Example 4.** For the example function $\kappa_{I4}$ (see Figure 1c), and from Example 2, and instance $((0, 0, 1, 1), 0)$, it becomes clear that feature 3 is irrelevant. Similarly, it is easy to conclude that features 1, 2 and 4 are relevant.

**How irrelevant are irrelevant features?** The fact that a feature is declared irrelevant for an explanation problem $\mathcal{E} = (\mathcal{M}, (\mathbf{v}, c))$ is significant. Given the minimal hitting set duality between abductive and contrastive explanations, then an irrelevant features does not occur neither in any abductive explanation, nor in any contrastive explanation. Furthermore, from the definition of AXp, each abductive explanation for $\mathcal{E}$ can be represented as a logic rule. Let $\mathcal{R}$ denote the set of *all irreducible* logic rules which can be used to predict $c$, given the literals dictated by $\mathbf{v}$. Then, an irrelevant feature does not occur in *any* of those rules. Example 4 illustrates the irrelevancy of feature 3, in that feature 3 would not occur in *any* irreducible rule for $\kappa_{I4}$ when predicting 0 using literals consistent with $(0, 0, 1, 1)$.

To further strengthen the above discussion, let us consider a (feature selection based) explanation $\mathcal{X} \subseteq \mathcal{F}$ such that WAXp($\mathcal{X}$) holds (i.e. (4) is true, and so $\mathcal{X}$ is sufficient for the prediction). Moreover, let $i \in \mathcal{F}$ be an irrelevant feature, such that $i \in \mathcal{X}$. Then, by definition of irrelevant feature, there *must* exist some $\mathcal{Z} \subseteq (\mathcal{X} \setminus \{i\})$, such that WAXp($\mathcal{Z}$) also holds (i.e. $\mathcal{Z}$ is *also* sufficient for the prediction). It is simple to understand why such set $\mathcal{Z}$ must exist. By definition of irrelevant feature, and because $i \in \mathcal{X}$, then $\mathcal{X}$ is not an AXp. However, there must exist an AXp $\mathcal{W} \subsetneq \mathcal{X}$ which, by definition of irrelevant feature, must not include $i$. Furthermore, and invoking Occam's razor[4], there is no reason to select $\mathcal{X}$ over $\mathcal{Z}$, and this remark applies to *any* set of features containing some irrelevant feature.

**Related work.** Shapley values for explainability is one of the hallmarks of feature attribution methods in XAI [64, 65, 20, 48, 15, 47, 52, 17, 26, 16, 25, 62, 40, 58, 69, 5, 12, 30, 4, 67]. Motivated by the success of Shapley values for explainability, there exists a burgeoning body of work on using Shapley values for explainability (e.g. [39, 74, 71, 38, 54, 10, 6, 76, 44, 3, 63, 75, 49, 68, 45, 46, 77, 28, 29, 31, 1]). Recent work studied the complexity of exactly computing Shapley values in the context of explainability [8, 21, 22]. Finally, there have been proposals for the exact computation of Shapley values in the case of circuit-based classifiers [8]. Although there exist some differences in the proposals for the use of Shapley values for explainability, the basic formulation is the same and can be expressed as in Section 2.

A number of authors have reported pitfalls with the use of SHAP and Shapley values as a measure of feature importance [73, 42, 66, 52, 27, 72, 55, 2, 70, 41, 14]. However, these earlier works do not identify fundamental flaws with the use of Shapley values in explainability. Attempts at addressing those pitfalls include proposals to integrate Shapley values with abductive explanations, as reported in recent work [43].

Formal explainability is a fairly recent topic of research. Recent accounts include [51, 9, 50, 19].

---

[3] All proofs are included in Appendix A.

[4] Here, we adopt a fairly standard definition of Occam's razor [11]: *given two explanations of the data, all other things being equal, the simpler explanation is preferable*.

Recent work [35] argued for the inadequacy of Shapley values for explainability, by demonstrating experimentally that the information provided by Shapley values can be misleading for a human decision-maker. The approach proposed in [35] is based on exhaustive function enumeration, and so does not scale beyond a few features. However, this paper uses the truth-table algorithms outlined in [35], in all the examples, both for computing Shapley values, for computing explanations, and for deciding feature relevancy.

## 3   Relating Shapley Values with Feature Relevancy

Recent work [35] showed the existence of boolean functions (with up to four variables) that revealed a number of issues with Shapley values for explainability. All those issues are related with taking feature relevancy into consideration. (In [35], these functions were searched by exhaustive enumeration of all the boolean functions up to a threshold on the number of variables.)

**Issues with Shapley values for explainability.** In this paper, we consider the following main issues of Shapley values for explainability:

**I1.** For a boolean classifier, with an instance $(\mathbf{v}, c)$, and feature $i$ such that,

$$\mathsf{Irrelevant}(i) \wedge (\mathsf{Sv}(i) \neq 0)$$

   Thus, an I1 issue is such that the feature is irrelevant, but its Shapley value is non-zero.

**I2.** For a boolean classifier, with an instance $(\mathbf{v}, c)$ and features $i_1$ and $i_2$ such that,

$$\mathsf{Irrelevant}(i_1) \wedge \mathsf{Relevant}(i_2) \wedge (|\mathsf{Sv}(i_1)| > |\mathsf{Sv}(i_2)|)$$

   Thus, an I2 issue is such that there is at least one irrelevant feature exhibiting a Shapley value larger (in absolute value) than the Shapley of a relevant feature.

**I3.** For a boolean classifier, with instance $(\mathbf{v}, c)$, and feature $i$ such that,

$$\mathsf{Relevant}(i) \wedge (\mathsf{Sv}(i) = 0)$$

   Thus, an I3 issue is such that the feature is relevant, but its Shapley value is zero.

**I4.** For a boolean classifier, with instance $(\mathbf{v}, c)$, and features $i_1$ and $i_2$ such that,

$$[\mathsf{Irrelevant}(i_1) \wedge (\mathsf{Sv}(i_1) \neq 0)] \wedge [\mathsf{Relevant}(i_2) \wedge (\mathsf{Sv}(i_2) = 0)]$$

   Thus, an I4 issue is such that there is at least one irrelevant feature with a non-zero Shapley value and a relevant feature with a Shapley value of 0.

**I5.** For a boolean classifier, with instance $(\mathbf{v}, c)$ and feature $i$ such that,

$$[\mathsf{Irrelevant}(i) \wedge \forall_{1 \leq j \leq m, j \neq i} (|\mathsf{Sv}(j)| < |\mathsf{Sv}(i)|)]$$

   Thus, an I5 issue is such that there is one irrelevant feature exhibiting the highest Shapley value (in absolute value). (I5 can be viewed as a special case of the other issues, and so it is not analyzed separately in earlier work [35].)

The issues above are all related with Shapley values for explainability giving *misleading information* to a human decision maker, by assigning some importance to irrelevant features, by not assigning enough importance to relevant features, by assigning more importance to irrelevant features than to relevant features and, finally, by assigning the most importance to irrelevant features.

In the rest of the paper we consider mostly I1, I3, I4 and I5, given that I5 implies I2.

**Proposition 2.** If a classifier and instance exhibits issue I5, then they also exhibit issue I2.

**Examples.** This section studies the example functions of Figure 1, which were derived from the main results of this paper (see Section 4). These example functions will then be used to motivate the rationale for how those results are proved. In all cases, the reported Shapley values are computed using the truth-table algorithm outlined in earlier work [35]. Similarly, the relevancy/irrelevancy claims of features use the truth-table algorithms outlined in earlier work [35].

**Example 5.** Figure 1a illustrates a boolean function that exhibits issue I1. By inspection, we can conclude that the function shown corresponds to $\kappa_{I1}(x_1, x_2, x_3) = (x_1 \wedge x_2 \wedge \neg x_3) \vee (x_1 \wedge x_3)$. Moreover, for the instance $((0, 0, 1), 0)$, Table 1 confirms that an issue I1 is identified.

**Example 6.** Figure 1b illustrates a boolean function that exhibits issue I3. By inspection, we can conclude that the function shown corresponds to $\kappa_{I3}(x_1, x_2, x_3) = (x_1 \wedge \neg x_3) \vee (x_2 \wedge x_3)$. Moreover, for the instance $((1, 1, 1), 1)$, Table 1 confirms that an issue I3 is identified.

Table 1: Examples of issues of Shapley values for functions in Figure 1

| Case | Instance | Relevant | Irrelevant | Sv's | Justification |
|------|----------|----------|------------|------|---------------|
| I1 | $((0,0,1),0)$ | 1 | 2,3 | $\mathsf{Sv}(1)=-0.417$ $\mathsf{Sv}(2)=-0.042$ $\mathsf{Sv}(3)=0.083$ | $\mathsf{Irrelevant}(3) \wedge \mathsf{Sv}(3) \neq 0$ |
| I3 | $((1,1,1),1)$ | 1,2,3 | – | $\mathsf{Sv}(1)=0.125$ $\mathsf{Sv}(2)=0.375$ $\mathsf{Sv}(3)=0.000$ | $\mathsf{Relevant}(3) \wedge \mathsf{Sv}(3) = 0$ |
| I4 | $((0,0,1,1),0)$ | 1,2,4 | 3 | $\mathsf{Sv}(1)=-0.125$ $\mathsf{Sv}(2)=-0.333$ $\mathsf{Sv}(3)=0.083$ $\mathsf{Sv}(4)=0.000$ | $\mathsf{Irrelevant}(3) \wedge \mathsf{Sv}(3) \neq 0 \wedge$ $\mathsf{Relevant}(4) \wedge \mathsf{Sv}(4) = 0$ |
| I5 | $((1,1,1,1),0)$ | 1,2,3 | 4 | $\mathsf{Sv}(1)=-0.12$ $\mathsf{Sv}(2)=-0.12$ $\mathsf{Sv}(3)=-0.12$ $\mathsf{Sv}(4)=0.17$ | $\mathsf{Irrelevant}(4) \wedge$ $\forall(j \in \{1,2,3\}).|\mathsf{Sv}(j)| < \mathsf{Sv}(4)|$ |

**Example 7.** Figure 1c illustrates a boolean function that exhibits issue I4. By inspection, we can conclude that the function shown corresponds to $\kappa_{I4}(x_1, x_2, x_3, x_4) = (x_1 \wedge x_2 \wedge \neg x_3) \vee (x_1 \wedge x_3 \wedge \neg x_4) \vee (x_2 \wedge x_3 \wedge x_4)$. Moreover, for the instance $((0,0,1,1),0)$, Table 1 confirms that an issue I4 is identified.

**Example 8.** Figure 1d illustrates a boolean function that exhibits issue I5. By inspection, we can conclude that the function shown corresponds to $\kappa_{I5}(x_1, x_2, x_3, x_4) = ((x_1 \wedge x_2 \wedge \neg x_3) \vee (x_1 \wedge x_3 \wedge \neg x_2) \vee (x_2 \wedge x_3 \wedge \neg x_1)) \wedge x_4$. Moreover, for the instance $((1,1,1,1),0)$, Table 1 confirms that an issue I5 is identified.

It should be underscored that Shapley values for explainability are *not* expected to give misleading information. Indeed, it is widely accepted that Shapley values measure the actual *influence* of a feature [64, 65, 48, 8, 21]. Concretely, [64] reads: "*...if a feature has no influence on the prediction it is assigned a contribution of 0.*" But [64] also reads "*According to the 2nd axiom, if two features values have an identical influence on the prediction they are assigned contributions of equal size. The 3rd axiom says that if a feature has no influence on the prediction it is assigned a contribution of 0.*" (In this last quote, the axioms refer to the axiomatic characterization of Shapley values.) Furthermore, one might be tempted to look at the value of the prediction and relate that with the computed Shapley value. For example, in the last row of Table 1, the prediction is 0, and the *irrelevant* feature 4 has a *positive* Shapley value. As a result, one might be tempted to believe that the irrelevant feature 4 would contribute to *changing* the value of the prediction. This is of course incorrect, since an irrelevant feature does not occur in *any* CXp's (besides not occurring in any AXp's) and so it is never necessary to changing the prediction. The key point here is that irrelevant features are *never* necessary, neither to keep nor to change the prediction.

## 4 Refuting Shapley Values for Explainability

The purpose of this section is to prove that for arbitrary large numbers of variables, there exist boolean functions and instances for which the Shapley values exhibit the issues reported in recent work [35], and detailed in Section 3. (Instead of detailed proofs, this section describes the key ideas of each proof. The detailed proofs are included in Appendix A.)

Throughout this section, let $m$ be the number of variables of the boolean functions we start from, and let $n$ denote the number of variables of the functions we will be constructing. In this case, we set $\mathcal{F} = \{1, \ldots, n\}$. Furthermore, for the sake of simplicity, we opt to introduce the new features as the last features (e.g., feature $n$). This choice does not affect the proof's argument in any way.

**Proposition 3.** For any $n \geq 3$, there exist boolean functions defined on $n$ variables, and at least one instance, which exhibit an issue I1, i.e. there exists an irrelevant feature $i \in \mathcal{F}$, such that $\mathsf{Sv}(i) \neq 0$.

*Proof idea.* The proof proposes to construct boolean functions, with an arbitrary number of variables (no smaller than 3), and the picking of an instance, such that a specific feature is irrelevant for the prediction, but its Shapley value is non-zero. To illustrate the construction, the example function from Figure 1a is used (see also Example 5).

The construction works as follows. We pick two non-constant functions $\kappa_1(x_1, \ldots, x_m)$ and $\kappa_2(x_1, \ldots, x_m)$, defined on $m$ features, and such that: i) $\kappa_1 \vDash \kappa_2$ (which signifies that $\forall(\mathbf{x} \in \mathbb{F}).\kappa_1(\mathbf{x}) \rightarrow \kappa_2(\mathbf{x})$), and ii) $\kappa_1 \neq \kappa_2$. Observe that $\kappa_1$ can be any boolean function defined on $m$ variables, as long as $\kappa_2$ can also be defined. We then construct a new function by adding a new feature $n = m + 1$, as follows:

$$\kappa(x_1, \ldots, x_m, x_n) = \begin{cases} \kappa_1(x_1, \ldots, x_m) & \text{if } x_n = 0 \\ \kappa_2(x_1, \ldots, x_m) & \text{if } x_n = 1 \end{cases}$$

For the resulting function $\kappa$, we pick an instance $(\mathbf{v}, 0)$ such that: i) $v_n = 1$ and ii) $\kappa_1(\mathbf{v}_{1..m}) = \kappa_2(\mathbf{v}_{1..m}) = 0$. The proof hinges on the fact that feature $n$ is irrelevant, but $\mathsf{Sv}(n) \neq 0$.

For the function Figure 1a, we set $\kappa_1(x_1, x_2) = x_1 \wedge x_2$ and $\kappa_1(x_1, x_2) = x_1$. Thus, as shown in Example 5, $\kappa_{I1}(x_1, x_2, x_3) = (x_1 \wedge x_2 \wedge \neg x_3) \vee (x_1 \wedge x_3)$, which represents the function $\kappa(x_1, x_2, x_3)$. It is also clear that $\kappa_1 \vDash \kappa_2$. Moreover, and as Example 5 and Table 1 show, it is the case that feature 3 is irrelevant and $\mathsf{Sv}(3) \neq 0$. $\qquad\square$

**Proposition 4.** For any odd $n \geq 3$, there exist boolean functions defined on $n$ variables, and at least one instance, which exhibits an I3 issue, i.e. for which there exists a relevant feature $i \in \mathcal{F}$, such that $\mathsf{Sv}(i) = 0$.

*Proof idea.* The proof proposes to construct boolean functions, with an arbitrary number of variables (no smaller than 3), and the picking of an instance, such that a specific feature is relevant for the prediction, but its Shapley value is zero. To illustrate the construction, the example function from Figure 1b is used (see also Example 6).

The construction works as follows. We pick two non-constant functions $\kappa_1(x_1, \ldots, x_m)$ and $\kappa_2(x_{m+1}, \ldots, x_{2m})$, each defined on $m$ features, where $\kappa_2$ corresponds to $\kappa_1$, but with a change of variables. Observe that $\kappa_1$ can be any boolean function. We then construct a new function, defined in terms of $\kappa_1$ and $\kappa_2$, by adding a new feature $n = 2m + 1$, as follows:

$$\kappa(x_1, \ldots, x_m, x_{m+1}, \ldots, x_{2m}, x_n) = \begin{cases} \kappa_1(x_1, \ldots, x_m) & \text{if } x_n = 0 \\ \kappa_2(x_{m+1}, \ldots, x_{2m}) & \text{if } x_n = 1 \end{cases}$$

For the resulting function $\kappa$, we pick an instance $(\mathbf{v}, 1)$ such that: i) $v_n = 1$, ii) $v_i = v_{m+i}$ for any $1 \leq i \leq m$, and iii) $\kappa_1(\mathbf{v}_{1..m}) = \kappa_2(\mathbf{v}_{m+1..2m}) = 1$. The proof hinges on the fact that feature $n$ is relevant, but $\mathsf{Sv}(n) = 0$.

For the function Figure 1b, we set $\kappa_1(x_1) = x_1$ and $\kappa_1(x_2) = x_2$. Thus, as shown in Example 6, $\kappa_{I3}(x_1, x_2, x_3) = (x_1 \wedge \neg x_3) \vee (\neg x_2 \wedge x_3)$, which represents the function $\kappa(x_1, x_2, x_3)$. Moreover, and as Example 6 and Table 1 show, it is the case that feature 3 is relevant and $\mathsf{Sv}(3) = 0$. $\qquad\square$

**Proposition 5.** For any even $n \geq 4$, there exist boolean functions defined on $n$ variables, and at least one instance, for which there exists an irrelevant feature $i_1 \in \mathcal{F}$, such that $\mathsf{Sv}(i_1) \neq 0$, and a relevant feature $i_2 \in \mathcal{F} \setminus \{i_1\}$, such that $\mathsf{Sv}(i_2) = 0$.

*Proof idea.* The proof proposes to construct boolean functions, with an arbitrary number of variables (no smaller than 4), and the picking of an instance, such that two specific features are such that one is relevant but has a Shapley value of 0, and the other one is irrelevant but has a non-zero Shapley values. To illustrate the construction, the example function from Figure 1c is used (see also Example 7).

The construction works as follows. We pick two non-constant functions $\kappa_1(x_1, \ldots, x_m)$ and $\kappa_2(x_{m+1}, \ldots, x_{2m})$, each defined on $m$ features, where $\kappa_2$ corresponds to $\kappa_1$, but with a change of variables. Also, observe that $\kappa_1$ can be any boolean function. We then construct a new function, defined in terms of $\kappa_1$ and $\kappa_2$, by adding two new features. We let the new features be $n - 1$ and $n$, and so $n = 2m + 2$. The function is organized as follows:

$$\kappa(\mathbf{x}_{1..m}, \mathbf{x}_{m+1..2m}, x_{n-1}, x_n) = \begin{cases} \kappa_1(\mathbf{x}_{1..m}) \wedge \kappa_2(\mathbf{x}_{m+1..2m}) & \text{if } x_{n-1} = 0 \\ \kappa_1(\mathbf{x}_{1..m}) & \text{if } x_{n-1} = 1 \wedge x_n = 0 \\ \kappa_2(\mathbf{x}_{m+1..2m}) & \text{if } x_{n-1} = 1 \wedge x_n = 1 \end{cases}$$

For this function, we pick an instance $(\mathbf{v}, 0)$ such that: i) $v_{n-1} = v_n = 1$, ii) $v_i = v_{m+i}$ for any $1 \leq i \leq m$, and iii) $\kappa_1(\mathbf{v}_{1..m}) = \kappa_2(\mathbf{v}_{m+1..2m}) = 0$. The proof hinges on the fact that feature $n - 1$ is irrelevant, feature $n$ is relevant, and $\mathsf{Sv}(n - 1) \neq 0$ and $\mathsf{Sv}(n) = 0$.

For the function Figure 1c, we set $\kappa_1(x_1) = x_1$ and $\kappa_1(x_2) = x_2$, Thus, as shown in Example 7, $\kappa_{I4}(x_1, x_2, x_3, x_4) = (x_1 \wedge x_2 \wedge \neg x_3) \vee (x_1 \wedge x_3 \wedge \neg x_4) \vee (x_2 \wedge x_3 \wedge x_4)$, which represents the function $\kappa(x_1, x_2, x_3, x_4)$. Moreover, and as Example 7 and Table 1 show, it is the case that feature 3 is irrelevant, feature 4 is relevant, and also $\mathsf{Sv}(3) \neq 0$ and $\mathsf{Sv}(4) = 0$. $\qquad\square$

**Proposition 6.** For any $n \geq 4$, there exists boolean functions defined on $n$ variables, and at least one instance, for which there exists an irrelevant feature $i \in \mathcal{F} = \{1, \ldots, n\}$, such that $|\mathsf{Sv}(i)| = \max\{|\mathsf{Sv}(j)| \,|\, j \in \mathcal{F}\}$.

*Proof idea.* The proof proposes to construct boolean functions, with an arbitrary number of variables (no smaller than 4), and the picking of an instance, such that one specific feature is irrelevant but it has the Shapley value with the largest absolute values. To illustrate the construction, the example function from Figure 1d is used (see also Example 8).

The construction works as follows. We pick one non-constant function $\kappa_1(x_1, \ldots, x_m)$, defined on $m$ features, such that: i) $\kappa_1$ predicts a specific point $\mathbf{v}_{1..m}$ as 0, moreover, for any point $\mathbf{x}_{1..m}$ such that $d_H(\mathbf{x}_{1..m}, \mathbf{v}_{1..m}) = 1$, $\kappa_1(\mathbf{x}_{1..m}) = 1$, where $d_H(\cdot)$ denotes the Hamming distance. ii) and $\kappa_1$ predicts all the other points as 0. For example, let $\kappa_1(x_1, \ldots, x_m) = 1$ iff $\sum_{i=1}^{m} \neg x_1 = 1$. We then construct a new function, defined in terms of $\kappa_1$, by adding one new feature. We let the new feature be $n$, and so $n = m + 1$. The new function is organized as follows:

$$\kappa(x_1, \ldots, x_m, x_n) = \begin{cases} 0 & \text{if } x_n = 0 \\ \kappa_1(x_1, \ldots, x_m) & \text{if } x_n = 1 \end{cases}$$

For this function, we pick the instance $(\mathbf{v}, 0)$ such that: i) $v_n = 1$, ii) $\mathbf{v}_{1..m}$ is the only point within the Hamming ball and iii) $\kappa_1(\mathbf{v}_{1..m}) = 0$. The proof hinges on the fact that feature $n$ is irrelevant, but $\forall(1 \leq j \leq m).|\mathsf{Sv}(j)| < |\mathsf{Sv}(n)|$.

For the function Figure 1d, we set $\kappa_1(x_1, x_2, x_3) = (x_1 \wedge x_2 \wedge \neg x_3) \vee (x_1 \wedge x_3 \wedge \neg x_2) \vee (x_2 \wedge x_3 \wedge \neg x_1)$ (i.e. the function takes value 1 when exactly one feature is 0). Thus, as shown in Example 7, $\kappa_{I5}(x_1, x_2, x_3, x_4) = ((x_1 \wedge x_2 \wedge \neg x_3) \vee (x_1 \wedge x_3 \wedge \neg x_2) \vee (x_2 \wedge x_3 \wedge \neg x_1)) \wedge x_4$, which represents the function $\kappa(x_1, x_2, x_3, x_4)$. Moreover, and as Example 8 and Table 1 show, it is the case that feature 4 is irrelevant and $\forall(1 \leq j \leq 3).|\mathsf{Sv}(j)| < |\mathsf{Sv}(4)|$. $\square$

For I2, we can restate the previous result, but such the functions constructed in the proof capture a more general family of functions.

**Proposition 7.** For any $n \geq 4$, there exist boolean functions defined on $n$ variables, and at least one instance, for which there exists an irrelevant feature $i_1 \in \mathcal{F}$, and a relevant feature $i_2 \in \mathcal{F} \setminus \{i_1\}$, such that $|\mathsf{Sv}(i_1)| > |\mathsf{Sv}(i_2)|$.

As noted above, for Propositions 3 to 5, the choice of the starting function is fairly flexible. In contrast, for Proposition 6, we pick *one* concrete function, which represents a trivial lower bound. As a result, and with the exception of I5, we can prove the following (fairly loose) lower bounds on the number of functions exhibiting the different issues.

**Proposition 8.** For Propositions 3 to 5,and Proposition 7 the following are lower bounds on the numbers issues exhibiting the respective issues:

1. For Proposition 3, a lower bound on the number of functions exhibiting I1 is $2^{2^{(n-1)}} - n - 3$.

2. For Proposition 4, a lower bound on the number of functions exhibiting I3 is $2^{2^{(n-1)/2}} - 2$.

3. For Proposition 5, a lower bound on the number of functions exhibiting I4 is $2^{2^{(n-2)/2}} - 2$.

4. For Proposition 7, a lower bound on the number of functions exhibiting I2 is $2^{2^{n-2} - (n-2) - 1} - 1$.

# 5 Conclusions

This paper gives theoretical arguments to the fact that Shapley values for explainability can produce misleading information about the relative importance of features. The paper distinguishes between the features that occur in one or more of the irreducible rule-based explanations, i.e. the *relevant* features, from those that do not occur in any irreducible rule-based explanation, i.e. the *irrelevant* features. The paper proves that, for boolean functions with arbitrary number of variables, irrelevant features can be deemed more important, given their Shapley value, than relevant features. Our results are also significant in practical deployment of explainability solutions. Indeed, misleading information about relative feature importance can induce human decision makers in error, by persuading them to look at the wrong causes of predictions.

One direction of research is to develop a better understanding of the distributions of functions exhibiting one or more of the issues of Shapley values.

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
