# A   Detailed Proofs

**Proposition 1.** $F_{\mathbb{A}}(\mathcal{E}) = F_{\mathbb{C}}(\mathcal{E})$.

*Proof.* This result is a consequence of minimal-hitting set duality between AXp's and CXp's, proved elsewhere [36]. □

**Proposition 2.** If a classifier and instance exhibits issue I5, then they also exhibit issue I2.

*Proof.* Given a classifier with classification function $\kappa$ and an instance $(\mathbf{v}, c)$, it is plain that the set of features $\mathcal{F}$ represents a WAXp. Furthermore, since the classification function is assumed not to be constant, then there must exist some AXp that is not the empty set. Thus, such AXp contains at least one relevant feature, say $i_{\mathrm{rel}} \in \mathcal{F}$. Moreover, if I5 holds, then there exists an irrelevant $i_{\mathrm{irr}} \in \mathcal{F} \setminus \{i_{\mathrm{rel}}\}$ with the largest absolute Shapley value. Therefore, it is the case that for feature $i_{\mathrm{rel}}$, its absolute Shapley value is smaller than that of irrelevant feature $i_{\mathrm{irr}}$. As a result, the function also exhibits issue I2. □

**Proposition 3.** For any $n \geq 3$, there exist boolean functions defined on $n$ variables, and at least one instance, which exhibit an issue I1, i.e. there exists an irrelevant feature $i \in \mathcal{F}$, such that $\mathsf{Sv}(i) \neq 0$.

*Proof.* Consider two classifiers $\mathcal{M}_1$ and $\mathcal{M}_2$ implementing non-constant boolean functions $\kappa_1$ and $\kappa_2$, respectively. These functions are defined on the set of features $\mathcal{F}' = \{1, \ldots, m\}$, and such that $\kappa_1 \models \kappa_2$ but $\kappa_1 \neq \kappa_2$. Consider the set of features $\mathcal{F} = \mathcal{F}' \cup \{n\}$, we construct a new classifier $\mathcal{M}$ by combining $\mathcal{M}_1$ and $\mathcal{M}_2$. The classifier $\mathcal{M}$ is characterized by the boolean function defined as follows:

$$\kappa(x_1, \ldots, x_m, x_n) := \begin{cases} \kappa_1(x_1, \ldots, x_m) & \text{if } x_n = 0 \\ \kappa_2(x_1, \ldots, x_m) & \text{if } x_n = 1 \end{cases} \tag{9}$$

Choose a $m$-dimensional point $\mathbf{v}_{1..m}$ such that $\kappa_1(\mathbf{v}_{1..m}) = \kappa_2(\mathbf{v}_{1..m}) = 0$, and extend $\mathbf{v}_{1..m}$ with $v_n = 1$. Then for the $n$-dimensional point $\mathbf{v}_{1..n} = (\mathbf{v}_{1..m}, 1)$, we have $\kappa(\mathbf{v}_{1..n}) = 0$.

To simplify the notation, we will use $\mathbf{x}'$ to denote an arbitrary $n$-dimensional point $\mathbf{x}_{1..n}$. Additionally, we will use $\mathbf{y}$ to denote an arbitrary $m$-dimensional point $\mathbf{x}_{1..m}$. For any subset $\mathcal{S} \subseteq \mathcal{F}'$, we have:

$$\phi(\mathcal{S} \cup \{n\}; \mathcal{M}, \mathbf{v}_{1..n}) - \phi(\mathcal{S}; \mathcal{M}, \mathbf{v}_{1..n}) \tag{10}$$

$$= \left( \frac{1}{2^{|(\mathcal{F}' \cup \{n\}) \setminus (\mathcal{S} \cup \{n\})|}} \sum_{\mathbf{x}' \in \Upsilon(\mathcal{S} \cup \{n\}; \mathbf{v}_{1..n})} \kappa(\mathbf{x}') \right) - \left( \frac{1}{2^{|(\mathcal{F}' \cup \{n\}) \setminus \mathcal{S}|}} \sum_{\mathbf{x}' \in \Upsilon(\mathcal{S}; \mathbf{v}_{1..n})} \kappa(\mathbf{x}') \right)$$

$$= \left( \frac{1}{2^{|\mathcal{F}' \setminus \mathcal{S}|}} \sum_{\mathbf{y} \in \Upsilon(\mathcal{S}; \mathbf{v}_{1..m})} \kappa_2(\mathbf{y}) \right) - \frac{1}{2^{|\mathcal{F}' \setminus \mathcal{S}|+1}} \left( \sum_{\mathbf{x}' \in \Upsilon(\mathcal{S};(\mathbf{v}_{1..m},1))} \kappa(\mathbf{x}') + \sum_{\mathbf{x}' \in \Upsilon(\mathcal{S};(\mathbf{v}_{1..m},0))} \kappa(\mathbf{x}') \right)$$

$$= \frac{1}{2^{|\mathcal{F}' \setminus \mathcal{S}|}} \left( \sum_{\mathbf{y} \in \Upsilon(\mathcal{S}; \mathbf{v}_{1..m})} \kappa_2(\mathbf{y}) - \frac{1}{2} \times \sum_{\mathbf{y} \in \Upsilon(\mathcal{S}; \mathbf{v}_{1..m})} \kappa_2(\mathbf{y}) - \frac{1}{2} \times \sum_{\mathbf{y} \in \Upsilon(\mathcal{S}; \mathbf{v}_{1..m})} \kappa_1(\mathbf{y}) \right)$$

$$= \frac{1}{2} \times \frac{1}{2^{|\mathcal{F}' \setminus \mathcal{S}|}} \left( \sum_{\mathbf{y} \in \Upsilon(\mathcal{S}; \mathbf{v}_{1..m})} \kappa_2(\mathbf{y}) - \sum_{\mathbf{y} \in \Upsilon(\mathcal{S}; \mathbf{v}_{1..m})} \kappa_1(\mathbf{y}) \right)$$

Given that $\kappa_1 \models \kappa_2$ but $\kappa_1 \neq \kappa_2$, it follows that for any points $\mathbf{y} \in \Upsilon(\mathcal{S}; \mathbf{v}_{1..m})$, if $\kappa_1(\mathbf{y}) = 1$ then $\kappa_2(\mathbf{y}) = 1$. In other words, if $\kappa_2(\mathbf{y}) = 0$ then $\kappa_1(\mathbf{y}) = 0$. Moreover, there are cases where the following inequality holds: $\sum_{\mathbf{y} \in \Upsilon(\mathcal{S}; \mathbf{v}_{1..m})} \kappa_2(\mathbf{y}) - \sum_{\mathbf{y} \in \Upsilon(\mathcal{S}; \mathbf{v}_{1..m})} \kappa_1(\mathbf{y}) > 0$. Hence, $\mathsf{Sv}(n) \neq 0$.

To prove that the feature $n$ is irrelevant, we assume the contrary, i.e., that $n$ is relevant, and $\mathcal{X}$ is an AXp of $\mathcal{M}$ for the point $\mathbf{v}_{1..n}$ such that $n \in \mathcal{X}$. This means we fix the variable $x_n$ to the value $v_n$, and, based on the definition of AXp, we only select the points that $\mathcal{M}_2$ predicts as 0. Since $\kappa_2(\mathbf{y}) = 0$ implies that $\kappa_1(\mathbf{y}) = 0$, removing feature $n$ from $\mathcal{X}$ means that $\mathcal{X} \setminus n$ will not include any points predicted as 1 by either $\mathcal{M}_1$ or $\mathcal{M}_2$. Thus, $\mathcal{X} \setminus n$ remains an AXp of $\mathcal{M}$ for the point $\mathbf{v}_{1..n}$, leading to a contradiction. Thus, feature $n$ is irrelevant. □

**Proposition 4.** For any odd $n \geq 3$, there exist boolean functions defined on $n$ variables, and at least one instance, which exhibits an I3 issue, i.e. for which there exists a relevant feature $i \in \mathcal{F}$, such that $\mathsf{Sv}(i) = 0$.

*Proof.* Given a classifier $\mathcal{M}_1$ implementing a non-constant boolean function $\kappa_1$ defined on the set of features $\mathcal{F}_1 = \{1, \ldots, m\}$. We can replace each $x_i$ of $\kappa_1$ with a new variable $x_{m+i}$ to obtain a new

function $\kappa_2$, defined on a new set of features $\mathcal{F}_2 = \{m+1, \ldots, 2m\}$. Importantly, $\kappa_2$ is independent of $\kappa_1$ as $\mathcal{F}_1$ and $\mathcal{F}_2$ are disjoint. Let $\mathcal{F} = \mathcal{F}_1 \cup \mathcal{F}_2 \cup \{n\}$, we build a new classifier $\mathcal{M}$ characterized by the boolean function defined as follows:

$$\kappa(x_1, \ldots, x_m, x_{m+1}, \ldots, x_{2m}, x_n) := \begin{cases} \kappa_1(x_1, \ldots, x_m) & \text{if } x_n = 0 \\ \kappa_2(x_{m+1}, \ldots, x_{2m}) & \text{if } x_n = 1 \end{cases} \tag{11}$$

Choose $m$-dimensional points $\mathbf{v}_{1..m}$ and $\mathbf{v}_{m+1..2m}$ such that $v_i = v_{m+i}$ for any $1 \leq i \leq m$, and $\kappa_1(\mathbf{v}_{1..m}) = \kappa_2(\mathbf{v}_{m+1..2m}) = 1$. Let $\mathbf{v}_{1..n} = (\mathbf{v}_{1..m}, \mathbf{v}_{m+1..2m}, 1)$ be a $n$-dimensional point such that $\kappa(\mathbf{v}_{1..n}) = 1$. Moreover, let $\mathcal{F}' = \mathcal{F}_1 \cup \mathcal{F}_2$.

To simplify the notations, we will use $\mathbf{u}$ to denote $\mathbf{v}_{1..m}$ and $\mathbf{w}$ to denote $\mathbf{v}_{m+1..2m}$, furthermore, we will use $\mathbf{x}'$ to denote an arbitrary $n$-dimensional point $\mathbf{x}_{1..n}$, and $\mathbf{y}$ to denote an arbitrary $m$-dimensional point $\mathbf{x}_{1..m}$, and $\mathbf{z}$ to denote an arbitrary $m$-dimensional point $\mathbf{x}_{m+1..2m}$. For any subset $\mathcal{S} \subseteq \mathcal{F}'$, let $\{\mathcal{S}_1, \mathcal{S}_2\}$ be a partition of $\mathcal{S}$ such that $\mathcal{S}_1 \subseteq \mathcal{F}_1 \wedge \mathcal{S}_2 \subseteq \mathcal{F}_2$, then:

$$\phi(\mathcal{S} \cup \{n\}; \mathcal{M}, \mathbf{v}_{1..n}) - \phi(\mathcal{S}; \mathcal{M}, \mathbf{v}_{1..n}) \tag{12}$$

$$= \left( \frac{1}{2^{|(\mathcal{F}' \cup \{n\}) \setminus (\mathcal{S} \cup \{n\})|}} \sum_{\mathbf{x}' \in \Upsilon(\mathcal{S} \cup \{n\}; \mathbf{v}_{1..n})} \kappa(\mathbf{x}') \right) - \left( \frac{1}{2^{|(\mathcal{F}' \cup \{n\}) \setminus \mathcal{S}|}} \sum_{\mathbf{x}' \in \Upsilon(\mathcal{S}; \mathbf{v}_{1..n})} \kappa(\mathbf{x}') \right)$$

$$= \frac{1}{2^{|\mathcal{F}' \setminus \mathcal{S}|}} \left( \sum_{\mathbf{x}' \in \Upsilon(\mathcal{S};(\mathbf{u},\mathbf{w},1))} \kappa(\mathbf{x}') - \frac{1}{2} \times \sum_{\mathbf{x}' \in \Upsilon(\mathcal{S};(\mathbf{u},\mathbf{w},1))} \kappa(\mathbf{x}') - \frac{1}{2} \times \sum_{\mathbf{x}' \in \Upsilon(\mathcal{S};(\mathbf{u},\mathbf{w},0))} \kappa(\mathbf{x}') \right)$$

$$= \frac{1}{2} \times \frac{1}{2^{|\mathcal{F}' \setminus \mathcal{S}|}} \left( \sum_{\mathbf{x}' \in \Upsilon(\mathcal{S};(\mathbf{u},\mathbf{w},1))} \kappa(\mathbf{x}') - \sum_{\mathbf{x}' \in \Upsilon(\mathcal{S};(\mathbf{u},\mathbf{w},0))} \kappa(\mathbf{x}') \right)$$

$$= \frac{1}{2} \times \frac{1}{2^{|\mathcal{F}' \setminus \mathcal{S}|}} \left( 2^{|\mathcal{F}_1 \setminus \mathcal{S}_1|} \times \sum_{\mathbf{z} \in \Upsilon(\mathcal{S}_2;\mathbf{w})} \kappa_2(\mathbf{z}) - 2^{|\mathcal{F}_2 \setminus \mathcal{S}_2|} \times \sum_{\mathbf{y} \in \Upsilon(\mathcal{S}_1;\mathbf{u})} \kappa_1(\mathbf{y}) \right)$$

For any $\{\mathcal{S}_1, \mathcal{S}_2\}$, we can construct a unique new partition $\{\mathcal{S}_1', \mathcal{S}_2'\}$ by replacing any $i \in \mathcal{S}_1$ with $m + i$ and any $m + i \in \mathcal{S}_2$ with $i$. Let $\mathcal{S}' = \mathcal{S}_1' \cup \mathcal{S}_2'$, then we have:

$$\phi(\mathcal{S}' \cup \{n\}; \mathcal{M}, \mathbf{v}_{1..n}) - \phi(\mathcal{S}'; \mathcal{M}, \mathbf{v}_{1..n}) \tag{13}$$

$$= \frac{1}{2} \times \frac{1}{2^{|\mathcal{F}' \setminus \mathcal{S}'|}} \left( 2^{|\mathcal{F}_1 \setminus \mathcal{S}_2'|} \times \sum_{\mathbf{z} \in \Upsilon(\mathcal{S}_1';\mathbf{w})} \kappa_2(\mathbf{z}) - 2^{|\mathcal{F}_2 \setminus \mathcal{S}_1'|} \sum_{\mathbf{y} \in \Upsilon(\mathcal{S}_2';\mathbf{u})} \kappa_1(\mathbf{y}) \right)$$

Besides, we have:

$$2^{|\mathcal{F}_1 \setminus \mathcal{S}_1|} \times \sum_{\mathbf{z} \in \Upsilon(\mathcal{S}_2;\mathbf{z})} \kappa_2(\mathbf{z}) = 2^{|\mathcal{F}_2 \setminus \mathcal{S}_1'|} \sum_{\mathbf{y} \in \Upsilon(\mathcal{S}_2';\mathbf{u})} \kappa_1(\mathbf{y})$$

and

$$2^{|\mathcal{F}_2 \setminus \mathcal{S}_2|} \times \sum_{\mathbf{y} \in \Upsilon(\mathcal{S}_1;\mathbf{u})} \kappa_1(\mathbf{y}) = 2^{|\mathcal{F}_1 \setminus \mathcal{S}_2'|} \times \sum_{\mathbf{z} \in \Upsilon(\mathcal{S}_1';\mathbf{w})} \kappa_2(\mathbf{z})$$

which means:

$$\phi(\mathcal{S} \cup \{n\}; \mathcal{M}, \mathbf{v}_{1..n}) - \phi(\mathcal{S}; \mathcal{M}, \mathbf{v}_{1..n}) = -(\phi(\mathcal{S}' \cup \{n\}; \mathcal{M}, \mathbf{v}_{1..n}) - \phi(\mathcal{S}'; \mathcal{M}, \mathbf{v}_{1..n}))$$

note that $\frac{|\mathcal{S}|!(|\mathcal{F}| - |\mathcal{S}| - 1)!}{|\mathcal{F}|!} = \frac{|\mathcal{S}'|!(|\mathcal{F}| - |\mathcal{S}'| - 1)!}{|\mathcal{F}|!}$. Hence, for any subset $\mathcal{S}$, there is a unique subset $\mathcal{S}'$ that can cancel its effect, from which we can derive that $\mathsf{Sv}(n) = 0$. However, $n$ is a relevant feature. To find an AXp containing $n$, we remove all features in $\mathcal{F}_1$, and keep only feature $n$ along with all features in $\mathcal{F}_2$. This makes feature $n$ critical to the change in the prediction of $\mathcal{M}$. Next, we compute an AXp $\mathcal{X}$ of $\mathcal{M}_2$ under the point $\mathbf{v}_{m+1..2m}$. Finally, $\mathcal{X} \cup \{n\}$ is an AXp of the classifier $\mathcal{M}$ for the point $\mathbf{v}_{1..n}$. □

**Proposition 5.** For any even $n \geq 4$, there exist boolean functions defined on $n$ variables, and at least one instance, for which there exists an irrelevant feature $i_1 \in \mathcal{F}$, such that $\mathsf{Sv}(i_1) \neq 0$, and a relevant feature $i_2 \in \mathcal{F} \setminus \{i_1\}$, such that $\mathsf{Sv}(i_2) = 0$.

*Proof.* Given a classifier $\mathcal{M}_1$ implementing a non-constant boolean function $\kappa_1$ defined on the set of features $\mathcal{F}_1 = \{1, \ldots, m\}$, we can construct a new classifier $\mathcal{M}$ characterized by the boolean function defined as follows:

$$\kappa(\mathbf{x}_{1..m}, \mathbf{x}_{m+1..2m}, x_{n-1}, x_n) := \begin{cases} \kappa_1(\mathbf{x}_{1..m}) \wedge \kappa_2(\mathbf{x}_{m+1..2m}) & \text{if } x_{n-1} = 0 \\ \kappa_1(\mathbf{x}_{1..m}) & \text{if } x_{n-1} = 1 \wedge x_n = 0 \\ \kappa_2(\mathbf{x}_{m+1..2m}) & \text{if } x_{n-1} = 1 \wedge x_n = 1 \end{cases} \tag{14}$$

where function $\kappa_2$ is obtained by replacing every $x_i$ of $\kappa_1$ with a new variable $x_{m+i}$. $\kappa_2$ is defined on a new set of features $\mathcal{F}_2 = \{m + 1, \ldots, 2m\}$ and is independent of $\kappa_1$. Moreover, $\mathcal{M}$ is defined on the feature set $\mathcal{F} = \mathcal{F}_1 \cup \mathcal{F}_2 \cup \{n - 1, n\}$. Note that $\kappa_1 \wedge \kappa_2 \models (\neg x_n \wedge \kappa_1) \vee (x_n \wedge \kappa_2)$, this can be proved using the consensus theorem [5].

Choose $m$-dimensional points $\mathbf{v}_{1..m}$ and $\mathbf{v}_{m+1..2m}$ such that $v_i = v_{m+i}$ for any $1 \le i \le m$, and $\kappa_1(\mathbf{v}_{1..m}) = \kappa_2(\mathbf{v}_{m+1..2m}) = 0$. Let $\mathbf{v}_{1..n} = (\mathbf{v}_{1..m}, \mathbf{v}_{m+1..2m}, 1, 1)$ be a $n$-dimensional point such that $\kappa(\mathbf{v}_{1..n}) = 0$. Moreover, let $\mathcal{F}' = \mathcal{F}_1 \cup \mathcal{F}_2$.

To simplify the notations, we will use $\mathbf{u}$ to denote $\mathbf{v}_{1..m}$ and $\mathbf{w}$ to denote $\mathbf{v}_{m+1..2m}$, furthermore, we will use $\mathbf{x}'$ to denote an arbitrary $n$-dimensional point $\mathbf{x}_{1..n}$, and $\mathbf{y}$ to denote an arbitrary $m$-dimensional point $\mathbf{x}_{1..m}$, and $\mathbf{z}$ to denote an arbitrary $m$-dimensional point $\mathbf{x}_{m+1..2m}$.

According to the proof of Proposition 3, $\mathsf{Sv}(n - 1) \ne 0$ but feature $n - 1$ is irrelevant. Next, we show that $\mathsf{Sv}(n) = 0$ but the feature $n$ is relevant. For any subset $\mathcal{S} \subseteq \mathcal{F}'$, let $\{\mathcal{S}_1, \mathcal{S}_2\}$ be a partition of $\mathcal{S}$ such that $\mathcal{S}_1 \subseteq \mathcal{F}_1 \wedge \mathcal{S}_2 \subseteq \mathcal{F}_2$.

1. Consider any subset $\mathcal{S} \cup \{n - 1\}$, then:

$$\phi(\mathcal{S} \cup \{n - 1, n\}; \mathcal{M}, \mathbf{v}_{1..n}) - \phi(\mathcal{S} \cup \{n - 1\}; \mathcal{M}, \mathbf{v}_{1..n}) \tag{15}$$

$$= \left( \frac{1}{2^{|\mathcal{F}' \setminus \mathcal{S}|}} \sum_{\mathbf{x}' \in \Upsilon(\mathcal{S} \cup \{n-1,n\}; \mathbf{v}_{1..n})} \kappa(\mathbf{x}') \right) - \left( \frac{1}{2^{|\mathcal{F}' \setminus \mathcal{S}|+1}} \sum_{\mathbf{x}' \in \Upsilon(\mathcal{S} \cup \{n-1\}; \mathbf{v}_{1..n})} \kappa(\mathbf{x}') \right)$$

$$= \frac{1}{2} \times \frac{1}{2^{|\mathcal{F}' \setminus \mathcal{S}|}} \left( \sum_{\mathbf{x}' \in \Upsilon(\mathcal{S} \cup \{n-1,n\}; (\mathbf{u}, \mathbf{w}, 1, 1))} \kappa(\mathbf{x}') - \sum_{\mathbf{x}' \in \Upsilon(\mathcal{S} \cup \{n-1\}; (\mathbf{u}, \mathbf{w}, 1, 0))} \kappa(\mathbf{x}') \right)$$

$$= \frac{1}{2} \times \frac{1}{2^{|\mathcal{F}' \setminus \mathcal{S}|}} \left( 2^{|\mathcal{F}_1 \setminus \mathcal{S}_1|} \times \sum_{\mathbf{z} \in \Upsilon(\mathcal{S}_2; \mathbf{w})} \kappa_2(\mathbf{z}) - 2^{|\mathcal{F}_2 \setminus \mathcal{S}_2|} \times \sum_{\mathbf{y} \in \Upsilon(\mathcal{S}_1; \mathbf{u})} \kappa_1(\mathbf{y}) \right)$$

According to the proof of Proposition 4, there is a unique subset $\mathcal{S}'$ such that $|\mathcal{S}| = |\mathcal{S}'|$ and $\phi(\mathcal{S} \cup \{n - 1, n\}; \mathcal{M}, \mathbf{v}_{1..n}) - \phi(\mathcal{S} \cup \{n - 1\}; \mathcal{M}, \mathbf{v}_{1..n}) = -(\phi(\mathcal{S}' \cup \{n - 1, n\}; \mathcal{M}, \mathbf{v}_{1..n}) - \phi(\mathcal{S}' \cup \{n - 1\}; \mathcal{M}, \mathbf{v}_{1..n}))$.

---

[5]The consensus theorem is the identity $(x \wedge y) \vee (\neg x \wedge z) = (x \wedge y) \vee (\neg x \wedge z) \vee (y \wedge z)$, see [18] Chapter 3

648   2. Consider any subset $\mathcal{S} \subseteq \mathcal{F}'$, then:

$$\phi(\mathcal{S} \cup \{n\}; \mathcal{M}, \mathbf{v}_{1..n}) - \phi(\mathcal{S}; \mathcal{M}, \mathbf{v}_{1..n}) \tag{16}$$

$$= \left( \frac{1}{2^{|\mathcal{F}' \setminus \mathcal{S}|+1}} \sum_{\mathbf{x}' \in \Upsilon(\mathcal{S} \cup \{n\}; \mathbf{v}_{1..n})} \kappa(\mathbf{x}') \right) - \left( \frac{1}{2^{|\mathcal{F}' \setminus \mathcal{S}|+2}} \sum_{\mathbf{x}' \in \Upsilon(\mathcal{S}; \mathbf{v}_{1..n})} \kappa(\mathbf{x}') \right)$$

$$= \frac{1}{2^{|\mathcal{F}' \setminus \mathcal{S}|+1}} \left( \sum_{\mathbf{x}' \in \Upsilon(\mathcal{S} \cup \{n\}; (\mathbf{u}, \mathbf{w}, 1, 1))} \kappa(\mathbf{x}') + \sum_{\mathbf{x}' \in \Upsilon(\mathcal{S} \cup \{n\}; (\mathbf{u}, \mathbf{w}, 0, 1))} \kappa(\mathbf{x}') \right)$$

$$- \frac{1}{2^{|\mathcal{F}' \setminus \mathcal{S}|+2}} \left( \sum_{\mathbf{x}' \in \Upsilon(\mathcal{S}; (\mathbf{u}, \mathbf{w}, 1, 1))} \kappa(\mathbf{x}') + \sum_{\mathbf{x}' \in \Upsilon(\mathcal{S}; (\mathbf{u}, \mathbf{w}, 0, 1))} \kappa(\mathbf{x}') \right)$$

$$- \frac{1}{2^{|\mathcal{F}' \setminus \mathcal{S}|+2}} \left( \sum_{\mathbf{x}' \in \Upsilon(\mathcal{S}; (\mathbf{u}, \mathbf{w}, 1, 0))} \kappa(\mathbf{x}') + \sum_{\mathbf{x}' \in \Upsilon(\mathcal{S}; (\mathbf{u}, \mathbf{w}, 0, 0))} \kappa(\mathbf{x}') \right)$$

$$= \frac{1}{4} \times \frac{1}{2^{|\mathcal{F}' \setminus \mathcal{S}|}} \left( 2^{|\mathcal{F}_1 \setminus \mathcal{S}_1|} \times \sum_{\mathbf{z} \in \Upsilon(\mathcal{S}_2; \mathbf{w})} \kappa_2(\mathbf{z}) - 2^{|\mathcal{F}_2 \setminus \mathcal{S}_2|} \times \sum_{\mathbf{y} \in \Upsilon(\mathcal{S}_1; \mathbf{u})} \kappa_1(\mathbf{y}) \right)$$

649   Likewise, we can find a unique subset $\mathcal{S}'$ to cancel the effect of $\phi(\mathcal{S} \cup \{n\}; \mathcal{M}, \mathbf{v}_{1..n}) -$
650   $\phi(\mathcal{S}; \mathcal{M}, \mathbf{v}1..n)$.

651   Therefore, $\mathsf{Sv}(n) = 0$. To prove that the feature $n$ is relevant, we compute an AXp containing the
652   feature $n$. First, we free all features in $\mathcal{F}_1$ and the feature $n - 1$, while keeping all features in $\mathcal{F}_2$ and
653   the feature $n$. This makes feature $n$ critical to the change in the prediction of $\mathcal{M}$. Next, we compute
654   an AXp $\mathcal{X}$ of $\mathcal{M}_2$ under the point $\mathbf{v}_{m+1..2m}$. Finally, we can conclude that $\mathcal{X} \cup \{n\}$ is an AXp of
655   $\mathcal{M}$ under the point $\mathbf{v}_{1..n}$. $\square$

656   **Proposition 6.** For any $n \geq 4$, there exists boolean functions defined on $n$ variables, and at least
657   one instance, for which there exists an irrelevant feature $i \in \mathcal{F} = \{1, \ldots, n\}$, such that $|\mathsf{Sv}(i)| =$
658   $\max\{|\mathsf{Sv}(j)| \mid j \in \mathcal{F}\}$.

659   *Proof.* Given a classifier $\mathcal{M}_1$ implementing a non-constant boolean function $\kappa_1$ defined on the set of
660   variables $\mathcal{F}' = \{1, \ldots, m\}$ where $m \geq 3$, and satisfies the following conditions:

661   1. $\kappa_1$ predicts a specific point $\mathbf{v}_{1..m}$ as 0. Furthermore, for any point $\mathbf{x}_{1..m}$ such that
662      $d_H(\mathbf{x}_{1..m}, \mathbf{v}_{1..m}) = 1$, where $d_H(\cdot)$ denotes the Hamming distance, we have $\kappa_1(\mathbf{x}_{1..m}) = 1$.
663   2. $\kappa_1$ predicts all the other points as 0.

664   For example, $\kappa_1$ can be the function $\sum_{i=1}^{m} \neg x_1 = 1$, which predicts the point $\mathbf{1}_{1..m}$ as 0 and all points
665   around this point with a Hamming distance of 1 as 1. Based on $\kappa_1$, we can build a new classifier $\mathcal{M}$
666   characterized by the boolean function defined as follows:

$$\kappa(x_1, \ldots, x_m, x_n) := \begin{cases} 0 & \text{if } x_n = 0 \\ \kappa_1(x_1, \ldots, x_m) & \text{if } x_n = 1 \end{cases} \tag{17}$$

667   Select the $m$-dimensional point $\mathbf{v}_{1..m}$ from our Hamming ball such that $\kappa_1(\mathbf{v}_{1..m}) = 0$ (note that
668   only one such point exists), and extend $\mathbf{v}_{1..m}$ with $v_n = 1$. Then for the $n$-dimensional point
669   $\mathbf{v}_{1..n} = (\mathbf{v}_{1..m}, 1)$, we have $\kappa(\mathbf{v}_{1..n}) = 0$. Applying the same reasoning presented in the proof of
670   Proposition 3, we can deduce that feature $n$ is irrelevant.

671   For simplicity, we will use $\mathbf{x}'$ to denote an arbitrary $n$-dimensional point $\mathbf{x}_{1..n}$, and $\mathbf{y}$ to denote an
672   arbitrary $m$-dimensional point $\mathbf{x}_{1..m}$. More importantly, for $\kappa_1$ and any subset $\mathcal{S} \subseteq \mathcal{F}'$, we have:

$$\sum_{\mathbf{y} \in \Upsilon(\mathcal{S}; \mathbf{v}_{1..m})} \kappa_1(\mathbf{y}) = m - |\mathcal{S}|$$

673    1. For the feature $n$ and an arbitrary subset $\mathcal{S} \subseteq \mathcal{F}'$, we have:

$$\phi(\mathcal{S} \cup \{n\}; \mathcal{M}, \mathbf{v}_{1..n}) - \phi(\mathcal{S}; \mathcal{M}, \mathbf{v}_{1..n}) \tag{18}$$

$$= \frac{1}{2^{|(\mathcal{F}' \cup \{n\}) \backslash (\mathcal{S} \cup \{n\})|}} \sum_{\mathbf{x}' \in \Upsilon(\mathcal{S} \cup \{n\}; \mathbf{v}_{1..n})} \kappa(\mathbf{x}') - \frac{1}{2^{|(\mathcal{F}' \cup \{n\}) \backslash \mathcal{S}|}} \sum_{\mathbf{x}' \in \Upsilon(\mathcal{S}; \mathbf{v}_{1..n})} \kappa(\mathbf{x}')$$

$$= \frac{1}{2^{|\mathcal{F}' \backslash \mathcal{S}|}} \sum_{\mathbf{x}' \in \Upsilon(\mathcal{S} \cup \{n\}; \mathbf{v}_{1..n})} \kappa(\mathbf{x}') - \frac{1}{2^{|\mathcal{F}' \backslash \mathcal{S}| + 1}} \sum_{\mathbf{x}' \in \Upsilon(\mathcal{S}; \mathbf{v}_{1..n})} \kappa(\mathbf{x}')$$

$$= \frac{1}{2} \times \frac{1}{2^{|\mathcal{F}' \backslash \mathcal{S}|}} \sum_{\mathbf{y} \in \Upsilon(\mathcal{S}; \mathbf{v}_{1..m})} \kappa_1(\mathbf{y})$$

$$= \frac{1}{2} \phi(\mathcal{S}; \mathcal{M}_1, \mathbf{v}_{1..m})$$

$$= \frac{1}{2} \times \frac{m - |\mathcal{S}|}{2^{m - |\mathcal{S}|}}$$

674    This means $\mathsf{Sv}(n) > 0$. Besides, the unique minimal value of $\phi(\mathcal{S} \cup \{n\}; \mathcal{M}, \mathbf{v}_{1..n}) -$
675    $\phi(\mathcal{S}; \mathcal{M}, \mathbf{v}_{1..n})$ is 0 when $\mathcal{S} = \mathcal{F}'$.
676    2. For a feature $j \neq n$, consider an arbitrary subset $\mathcal{S} \subseteq \mathcal{F}' \backslash \{j\}$ and the feature $n$, we have:

$$\phi(\mathcal{S} \cup \{j, n\}; \mathcal{M}, \mathbf{v}_{1..n}) - \phi(\mathcal{S} \cup \{n\}; \mathcal{M}, \mathbf{v}_{1..n}) \tag{19}$$

$$= \frac{1}{2^{|(\mathcal{F}' \cup \{n\}) \backslash (\mathcal{S} \cup \{j, n\})|}} \sum_{\mathbf{x}' \in \Upsilon(\mathcal{S} \cup \{j, n\}; \mathbf{v}_{1..n})} \kappa(\mathbf{x}') - \frac{1}{2^{|(\mathcal{F}' \cup \{n\}) \backslash (\mathcal{S} \cup \{n\})|}} \sum_{\mathbf{x}' \in \Upsilon(\mathcal{S} \cup \{n\}; \mathbf{v}_{1..n})} \kappa(\mathbf{x}')$$

$$= \frac{1}{2^{|\mathcal{F}' \backslash (\mathcal{S} \cup \{j\})|}} \sum_{\mathbf{y} \in \Upsilon(\mathcal{S} \cup \{j\}; \mathbf{v}_{1..m})} \kappa_1(\mathbf{y}) - \frac{1}{2^{|\mathcal{F}' \backslash \mathcal{S}|}} \sum_{\mathbf{y} \in \Upsilon(\mathcal{S}; \mathbf{v}_{1..m})} \kappa_1(\mathbf{y})$$

$$= \phi(\mathcal{S} \cup \{j\}; \mathcal{M}_1, \mathbf{v}_{1..m}) - \phi(\mathcal{S}; \mathcal{M}_1, \mathbf{v}_{1..m})$$

$$= \frac{m - |\mathcal{S}| - 1}{2^{m - |\mathcal{S}| - 1}} - \frac{m - |\mathcal{S}|}{2^{m - |\mathcal{S}|}}$$

$$= \frac{m - |\mathcal{S}| - 2}{2^{m - |\mathcal{S}|}}$$

677    In this case, $\phi(\mathcal{S} \cup \{j, n\}; \mathcal{M}, \mathbf{v}_{1..n}) - \phi(\mathcal{S} \cup \{n\}; \mathcal{M}, \mathbf{v}_{1..n}) = -\frac{1}{2}$ if $|\mathcal{S}| = m - 1$, which is its
678    unique minimal value. $\phi(\mathcal{S} \cup \{j, n\}; \mathcal{M}, \mathbf{v}_{1..n}) - \phi(\mathcal{S} \cup \{n\}; \mathcal{M}, \mathbf{v}_{1..n}) = 0$ if $|\mathcal{S}| = m - 2$, and
679    $\phi(\mathcal{S} \cup \{j, n\}; \mathcal{M}, \mathbf{v}_{1..n}) - \phi(\mathcal{S} \cup \{n\}; \mathcal{M}, \mathbf{v}_{1..n}) > 0$ if $|\mathcal{S}| < m - 2$.
680    3. Moreover, for a feature $j \neq n$, consider an arbitrary subset $\mathcal{S} \subseteq \mathcal{F}' \backslash \{j\}$ and without the feature
681    $n$, we have:

$$\phi(\mathcal{S} \cup \{j\}; \mathcal{M}, \mathbf{v}_{1..n}) - \phi(\mathcal{S}; \mathcal{M}, \mathbf{v}_{1..n}) \tag{20}$$

$$= \frac{1}{2^{|(\mathcal{F}' \cup \{n\}) \backslash (\mathcal{S} \cup \{j\})|}} \sum_{\mathbf{x}' \in \Upsilon(\mathcal{S} \cup \{j\}; \mathbf{v}_{1..n})} \kappa(\mathbf{x}') - \frac{1}{2^{|(\mathcal{F}' \cup \{n\}) \backslash \mathcal{S}|}} \sum_{\mathbf{x}' \in \Upsilon(\mathcal{S}; \mathbf{v}_{1..n})} \kappa(\mathbf{x}')$$

$$= \frac{1}{2^{|\mathcal{F}' \backslash (\mathcal{S} \cup \{j\})| + 1}} \sum_{\mathbf{y} \in \Upsilon(\mathcal{S} \cup \{j\}; \mathbf{v}_{1..m})} \kappa_1(\mathbf{y}) - \frac{1}{2^{|\mathcal{F}' \backslash \mathcal{S}| + 1}} \sum_{\mathbf{y} \in \Upsilon(\mathcal{S}; \mathbf{v}_{1..m})} \kappa_1(\mathbf{y})$$

$$= \frac{1}{2} (\phi(\mathcal{S} \cup \{j\}; \mathcal{M}_1, \mathbf{v}_{1..m}) - \phi(\mathcal{S}; \mathcal{M}_1, \mathbf{v}_{1..m}))$$

$$= \frac{1}{2} \times \frac{m - |\mathcal{S}| - 2}{2^{m - |\mathcal{S}|}}$$

682    In this case, $\phi(\mathcal{S} \cup \{j\}; \mathcal{M}, \mathbf{v}_{1..n}) - \phi(\mathcal{S}; \mathcal{M}, \mathbf{v}_{1..n}) = -\frac{1}{4}$ if $|\mathcal{S}| = m - 1$, which is its
683    unique minimal value. $\phi(\mathcal{S} \cup \{j\}; \mathcal{M}, \mathbf{v}_{1..n}) - \phi(\mathcal{S}; \mathcal{M}, \mathbf{v}_{1..n}) = 0$ if $|\mathcal{S}| = m - 2$, and
684    $\phi(\mathcal{S} \cup \{j\}; \mathcal{M}, \mathbf{v}_{1..n}) - \phi(\mathcal{S}; \mathcal{M}, \mathbf{v}_{1..n}) > 0$ if $|\mathcal{S}| < m - 2$.

685    Next, we prove $|\mathsf{Sv}(n)| > |\mathsf{Sv}(j)|$ by showing $\mathsf{Sv}(n) + \mathsf{Sv}(j) > 0$ and $\mathsf{Sv}(n) - \mathsf{Sv}(j) > 0$. Note
686    that $\mathsf{Sv}(n) > 0$. Additionally, $\phi(\mathcal{S} \cup \{j, n\}; \mathcal{M}, \mathbf{v}_{1..n}) - \phi(\mathcal{S} \cup \{n\}; \mathcal{M}, \mathbf{v}_{1..n}) < 0$ and $\phi(\mathcal{S} \cup$
687    $\{j\}; \mathcal{M}, \mathbf{v}_{1..n}) - \phi(\mathcal{S}; \mathcal{M}, \mathbf{v}_{1..n}) < 0$ only when $|\mathcal{S}| = m - 1$.

1. For $\mathsf{Sv}(n)$:

$$\mathsf{Sv}(n) = \sum_{\mathcal{S} \subseteq \mathcal{F} \setminus \{n\}} \frac{|\mathcal{S}|!(m - |\mathcal{S}|)!}{(m+1)!} \times (\phi(\mathcal{S} \cup \{n\}; \mathcal{M}, \mathbf{v}_{1..n}) - \phi(\mathcal{S}; \mathcal{M}, \mathbf{v}_{1..n})) \qquad (21)$$

$$= \sum_{\mathcal{S} \subseteq \mathcal{F} \setminus \{n\}} \frac{|\mathcal{S}|!(m - |\mathcal{S}|)!}{(m+1)!} \times \frac{1}{2} \phi(\mathcal{S}; \mathcal{M}_1, \mathbf{v}_{1..m})$$

$$= \frac{1}{2} \times \frac{1}{m+1} \times \sum_{\mathcal{S} \subseteq \mathcal{F} \setminus \{n\}} \frac{|\mathcal{S}|!(m - |\mathcal{S}|)!}{m!} \phi(\mathcal{S}; \mathcal{M}_1, \mathbf{v}_{1..m})$$

$$= \frac{1}{2} \times \frac{1}{m+1} \times \sum_{\mathcal{S} \subseteq \mathcal{F} \setminus \{n\}} \frac{|\mathcal{S}|!(m - |\mathcal{S}|)!}{m!} \times \frac{m - |\mathcal{S}|}{2^{m-|\mathcal{S}|}}$$

$$= \frac{1}{2} \times \frac{1}{m+1} \times \sum_{0 \le |\mathcal{S}| \le m} \frac{|\mathcal{S}|!(m - |\mathcal{S}|)!}{m!} \times \frac{m!}{|\mathcal{S}|!(m - |\mathcal{S}|)!} \times \frac{m - |\mathcal{S}|}{2^{m-|\mathcal{S}|}}$$

$$= \frac{1}{2} \times \frac{1}{m+1} \times \sum_{0 \le |\mathcal{S}| \le m} \frac{m - |\mathcal{S}|}{2^{m-|\mathcal{S}|}} = \frac{1}{2} \times \frac{1}{m+1} \times \sum_{k=1}^{m} \frac{k}{2^k}$$

$$= \frac{1}{2} \times \frac{1}{m+1} \times \frac{2^{m+1} - m - 2}{2^m} = \frac{1}{m+1} \times \frac{2^{m+1} - m - 2}{2^{m+1}}$$

2. For a feature $j \ne n$, consider the subset $\mathcal{S} = \mathcal{F}' \setminus \{j\}$ where $|\mathcal{S}| = m - 1$ and the feature $n$:

$$\frac{|\mathcal{S} \cup \{n\}|!(m - |\mathcal{S} \cup \{n\}|)!}{(m+1)!} \times \frac{m - |\mathcal{S}| - 2}{2^{m-|\mathcal{S}|}} \qquad (22)$$

$$= \frac{m!(m - m)!}{(m+1)!} \times \frac{m - (m-1) - 2}{2^{m-(m-1)}}$$

$$= -\frac{1}{2} \times \frac{1}{m+1}$$

3. For a feature $j \ne n$, consider the subset $\mathcal{S} = \mathcal{F}' \setminus \{j\}$ where $|\mathcal{S}| = m - 1$ and without the feature $n$:

$$\frac{|\mathcal{S}|!(m - |\mathcal{S}|)!}{(m+1)!} \times \frac{1}{2} \times \frac{m - |\mathcal{S}| - 2}{2^{m-|\mathcal{S}|}} \qquad (23)$$

$$= \frac{1}{2} \times \frac{(m-1)!(m - (m-1))!}{(m+1)!} \times \frac{m - (m-1) - 2}{2^{m-(m-1)}}$$

$$= -\frac{1}{4} \times \frac{1}{m(m+1)}$$

We consider the sum of these three values:

$$\frac{1}{m+1} \times \frac{2^{m+1} - m - 2}{2^{m+1}} - \frac{1}{2} \times \frac{1}{m+1} - \frac{1}{4} \times \frac{1}{m(m+1)} \qquad (24)$$

$$= \frac{1}{m+1} \times \left( \frac{(2^{m+1} - m - 2)m}{m 2^{m+1}} - \frac{m 2^m}{m 2^{m+1}} - \frac{2^{m-1}}{m 2^{m+1}} \right)$$

$$= \frac{1}{m(m+1)2^{m+1}} \times \left( (m - \frac{1}{2})2^m - m^2 - 2m \right)$$

Since $m \ge 3$, the sum of these three values is always greater than 0. Thus, we can conclude that $\mathsf{Sv}(n) + \mathsf{Sv}(j) > 0$.

To show $\mathsf{Sv}(n) - \mathsf{Sv}(j) > 0$, we focus on all subsets $\mathcal{S} \subseteq \mathcal{F}'$ where $|\mathcal{S}| < m - 2$. This is because, as previously stated, $\phi(\mathcal{S} \cup \{j, n\}; \mathcal{M}, \mathbf{v}_{1..n}) - \phi(\mathcal{S} \cup \{n\}; \mathcal{M}, \mathbf{v}_{1..n}) \le 0$ and $\phi(\mathcal{S} \cup \{j\}; \mathcal{M}, \mathbf{v}_{1..n}) - \phi(\mathcal{S}; \mathcal{M}, \mathbf{v}_{1..n}) \le 0$ if $|\mathcal{S}| \ge m - 2$.

Moreover, for all subsets $\mathcal{S} \subseteq \mathcal{F}'$ where $|\mathcal{S}| = k$ where $0 < k \le m - 3$, we compute the following three quantities:

$$Q_1 := \sum_{\mathcal{S} \subseteq \mathcal{F}', |\mathcal{S}| = k} \phi(\mathcal{S} \cup \{n\}; \mathcal{M}, \mathbf{v}_{1..n}) - \phi(\mathcal{S}; \mathcal{M}, \mathbf{v}_{1..n})$$

$$Q_2 := \sum_{\mathcal{S} \subseteq \mathcal{F}' \setminus \{j\}, |\mathcal{S}| = k-1} \phi(\mathcal{S} \cup \{j, n\}; \mathcal{M}, \mathbf{v}_{1..n}) - \phi(\mathcal{S} \cup \{n\}; \mathcal{M}, \mathbf{v}_{1..n})$$

$$Q_3 := \sum_{\mathcal{S} \subseteq \mathcal{F}' \setminus \{j\}, |\mathcal{S}| = k} \phi(\mathcal{S} \cup \{j\}; \mathcal{M}, \mathbf{v}_{1..n}) - \phi(\mathcal{S}; \mathcal{M}, \mathbf{v}_{1..n})$$

and show that $Q_1 - Q_2 - Q_3 > 0$. Note that $Q_1$, $Q_2$ and $Q_3$ share the same coefficient $\frac{k!(n-k-1)!}{n!}$.

1. For the feature $n$, we pick all possible subsets $\mathcal{S} \subseteq \mathcal{F}'$ where $|\mathcal{S}| = k$, which implies $|\mathcal{S} \cup \{n\}| = k + 1$, then:

$$Q_1 = \binom{m}{|\mathcal{S}|} \times \frac{1}{2} \times \frac{m - |\mathcal{S}|}{2^{m-|\mathcal{S}|}} = \binom{m}{k} \times \frac{1}{2} \times \frac{m - k}{2^{m-k}}$$

2. For a feature $j \neq n$ and consider the feature $n$, we pick all possible subsets $\mathcal{S} \subseteq \mathcal{F}'$ where $|\mathcal{S}| = k - 1$, which implies $|\mathcal{S} \cup \{j, n\}| = k + 1$, then:

$$Q_2 = \binom{m-1}{|\mathcal{S}|} \times \frac{m - |\mathcal{S}| - 2}{2^{m-|\mathcal{S}|}} = \binom{m-1}{k-1} \times \frac{m - (k-1) - 2}{2^{m-(k-1)}} = \binom{m-1}{k-1} \times \frac{1}{2} \times \frac{m - k - 1}{2^{m-k}}$$

3. For a feature $j \neq n$, without considering the feature $n$, we pick all possible subsets $\mathcal{S} \subseteq \mathcal{F}'$ where $|\mathcal{S}| = k$, which implies $|\mathcal{S} \cup \{j\}| = k + 1$, then:

$$Q_3 = \binom{m-1}{|\mathcal{S}|} \times \frac{1}{2} \times \frac{m - |\mathcal{S}| - 2}{2^{m-|\mathcal{S}|}} = \binom{m-1}{k} \times \frac{1}{2} \times \frac{m - k - 2}{2^{m-k}}$$

Then we compute $Q_1 - Q_2 - Q_3$:

$$\binom{m}{k} \times \frac{1}{2} \times \frac{m - k}{2^{m-k}} - \binom{m-1}{k-1} \times \frac{1}{2} \times \frac{m - k - 1}{2^{m-k}} - \binom{m-1}{k} \times \frac{1}{2} \times \frac{m - k - 2}{2^{m-k}} \tag{25}$$

$$= \frac{1}{2} \times \frac{1}{2^{m-k}} \left[ \binom{m}{k}(m-k) - \binom{m-1}{k-1}(m-k-1) - \binom{m-1}{k}(m-k-2) \right]$$

$$= \frac{1}{2} \times \frac{1}{2^{m-k}} \left[ \binom{m}{k}(m-k) - \binom{m-1}{k-1}(m-k) - \binom{m-1}{k}(m-k) + \binom{m-1}{k-1} + 2\binom{m-1}{k} \right]$$

$$= \frac{1}{2} \times \frac{1}{2^{m-k}} \left[ \binom{m-1}{k-1} + 2\binom{m-1}{k} \right]$$

This means that $\mathsf{Sv}(n) - \mathsf{Sv}(j) > 0$. Hence, we can conclude that $|\mathsf{Sv}(n)| > |\mathsf{Sv}(j)|$. $\qquad \square$

**Proposition 7.** For any $n \geq 4$, there exist boolean functions defined on $n$ variables, and at least one instance, for which there exists an irrelevant feature $i_1 \in \mathcal{F}$, and a relevant feature $i_2 \in \mathcal{F} \setminus \{i_1\}$, such that $|\mathsf{Sv}(i_1)| > |\mathsf{Sv}(i_2)|$.

*Proof.* Consider three classifiers $\mathcal{M}_1$, $\mathcal{M}_2$ and $\mathcal{M}_3$ implementing non-constant boolean functions $\kappa_1$, $\kappa_2$ and $\kappa_3$, respectively. Actually it is possible for $\kappa_1$ to be the constant function 0. All of them are defined on the set of features $\mathcal{F}' = \{1, \ldots, m\}$ where $m \geq 2$. More importantly, $\kappa_1$, $\kappa_2$ and $\kappa_3$ satisfy the following conditions:

1. $\kappa_2$ is a function predicting exactly one point $\mathbf{v}_{1..m}$ to 1, for example, $\kappa_2$ can be $\bigwedge_{1 \leq i \leq m} \neg x_i$.
2. For the point $\mathbf{v}_{1..m}$ where $\kappa_2$ predicts 1, we have $\kappa_3(\mathbf{v}_{1..m}) = 0$. This implies $\kappa_2 \wedge \kappa_3 \models \bot$, that is, the conjunction of $\kappa_2$ and $\kappa_3$ is logically inconsistent.
3. For any point $\mathbf{x}_{1..m}$ such that $d_H(\mathbf{x}_{1..m}, \mathbf{v}_{1..m}) = 1$, where $d_H(\cdot)$ denotes the Hamming distance, we have $\kappa_3(\mathbf{x}_{1..m}) = 1$.
4. $\kappa_1 \wedge \kappa_2 \models \bot$ and $\kappa_1 \wedge \kappa_3 \models \bot$, indicating that the conjunction of $\kappa_1$ and $\kappa_2$ as well as the conjunction of $\kappa_1$ and $\kappa_3$ both equal to the constant function 0.
5. $\kappa_1 \vee \kappa_2 \neq 1$ and $\kappa_1 \vee \kappa_3 \neq 1$, indicating that neither the disjunction of $\kappa_1$ and $\kappa_2$ nor the disjunction of $\kappa_1$ and $\kappa_3$ equals the constant function 1.

Let $\mathcal{F} = \mathcal{F}' \cup \{n-1, n\}$, we can build a new classifier $\mathcal{M}$ from $\mathcal{M}_1$, $\mathcal{M}_2$ and $\mathcal{M}_3$. $\mathcal{M}$ is characterized by the boolean function defined as follows:

$$\kappa(\mathbf{x}_{1..m}, x_{n-1}, x_n) := \begin{cases} \kappa_1(\mathbf{x}_{1..m}) & \text{if } x_{n-1} = 0 \\ \kappa_1(\mathbf{x}_{1..m}) \vee \kappa_2(\mathbf{x}_{1..m}) & \text{if } x_{n-1} = 1 \wedge x_n = 0 \\ \kappa_1(\mathbf{x}_{1..m}) \vee \kappa_3(\mathbf{x}_{1..m}) & \text{if } x_{n-1} = 1 \wedge x_n = 1 \end{cases} \tag{26}$$

Besides, we can derive that $(\neg x_n \wedge (\kappa_1 \vee \kappa_2)) \vee (x_n \wedge (\kappa_1 \vee \kappa_3)) = \kappa_1 \vee (\neg x_n \wedge \kappa_2) \vee (x_n \wedge \kappa_3)$. So we have $\kappa_1 \models \kappa_1 \vee (\neg x_n \wedge \kappa_2) \vee (x_n \wedge \kappa_3)$. Choose the $m$-dimensional point $\mathbf{v}_{1..m}$ such that $\kappa_1(\mathbf{v}_{1..m}) = \kappa_3(\mathbf{v}_{1..m}) = 0$ but $\kappa_2(\mathbf{v}_{1..m}) = 1$. Extend $\mathbf{v}_{1..m}$ with $v_{n-1} = v_n = 1$, let $\mathbf{v}_{1..n} = (\mathbf{v}_{1..m}, 1, 1)$ be the $n$-dimensional point, it follows that $\kappa(\mathbf{v}_{1..n}) = 0$. Based on the proof of Proposition 3, feature $n - 1$ is irrelevant. To prove that feature $n$ is relevant, we assume the contrary, i.e., that $n$ is irrelevant. In this case, we pick the point $\mathbf{v}' = (\mathbf{v}_{1..m}, 1, 0)$ from the feature space where $\kappa_2(\mathbf{v}_{1..m}) = 1$. Clearly, for this point we have $\kappa(\mathbf{v}') = 1$, leading to a contradiction. Thus, feature $n$ is relevant.

In the following, we prove that $|\mathsf{Sv}(n - 1)| > |\mathsf{Sv}(n)|$ by showing that $\mathsf{Sv}(n - 1) - \mathsf{Sv}(n) > 0$ and $\mathsf{Sv}(n-1) + \mathsf{Sv}(n) > 0$. To simplify the notations, we will use $\mathbf{x}'$ to denote an arbitrary $n$-dimensional point $\mathbf{x}_{1..n}$, and $\mathbf{y}$ to denote an arbitrary $m$-dimensional point $\mathbf{x}_{1..m}$. For any subset $\mathcal{S} \subseteq \mathcal{F}'$, we now focus on feature $n - 1$.

1. For the feature $n - 1$, consider an arbitrary subset $\mathcal{S} \subseteq \mathcal{F}'$ and without the feature $n$, then:

$$
\phi(\mathcal{S} \cup \{n - 1\}; \mathcal{M}, \mathbf{v}_{1..n}) - \phi(\mathcal{S}; \mathcal{M}, \mathbf{v}_{1..n}) \tag{27}
$$

$$
= \left( \frac{1}{2^{|\mathcal{F}' \setminus \mathcal{S}|+1}} \sum_{\mathbf{x}' \in \Upsilon(\mathcal{S} \cup \{n-1\}; \mathbf{v}_{1..n})} \kappa(\mathbf{x}') \right) - \left( \frac{1}{2^{|\mathcal{F}' \setminus \mathcal{S}|+2}} \sum_{\mathbf{x}' \in \Upsilon(\mathcal{S}; \mathbf{v}_{1..n})} \kappa(\mathbf{x}') \right)
$$

$$
= \frac{1}{2^{|\mathcal{F}' \setminus \mathcal{S}|+1}} \left( \sum_{\mathbf{y} \in \Upsilon(\mathcal{S}; \mathbf{v}_{1..m})} (\kappa_1(\mathbf{y}) \vee \kappa_3(\mathbf{y})) + \sum_{\mathbf{y} \in \Upsilon(\mathcal{S}; \mathbf{v}_{1..m})} (\kappa_1(\mathbf{y}) \vee \kappa_2(\mathbf{y})) \right)
$$

$$
- \frac{1}{2^{|\mathcal{F}' \setminus \mathcal{S}|+2}} \left( \sum_{\mathbf{y} \in \Upsilon(\mathcal{S}; \mathbf{v}_{1..m})} (\kappa_1(\mathbf{y}) \vee \kappa_3(\mathbf{y})) + \sum_{\mathbf{y} \in \Upsilon(\mathcal{S}; \mathbf{v}_{1..m})} (\kappa_1(\mathbf{y}) \vee \kappa_2(\mathbf{y})) \right)
$$

$$
- \frac{1}{2^{|\mathcal{F}' \setminus \mathcal{S}|+2}} \left( \sum_{\mathbf{y} \in \Upsilon(\mathcal{S}; \mathbf{v}_{1..m})} \kappa_1(\mathbf{y}) + \sum_{\mathbf{y} \in \Upsilon(\mathcal{S}; \mathbf{v}_{1..m})} \kappa_1(\mathbf{y}) \right)
$$

$$
= \frac{1}{2^{|\mathcal{F}' \setminus \mathcal{S}|+2}} \left( \sum_{\mathbf{y} \in \Upsilon(\mathcal{S}; \mathbf{v}_{1..m})} (\kappa_1(\mathbf{y}) \vee \kappa_3(\mathbf{y})) + \sum_{\mathbf{y} \in \Upsilon(\mathcal{S}; \mathbf{v}_{1..m})} (\kappa_1(\mathbf{y}) \vee \kappa_2(\mathbf{y})) \right)
$$

$$
- \frac{1}{2^{|\mathcal{F}' \setminus \mathcal{S}|+2}} \left( \sum_{\mathbf{y} \in \Upsilon(\mathcal{S}; \mathbf{v}_{1..m})} \kappa_1(\mathbf{y}) + \sum_{\mathbf{y} \in \Upsilon(\mathcal{S}; \mathbf{v}_{1..m})} \kappa_1(\mathbf{y}) \right)
$$

$$
= \frac{1}{2^{|\mathcal{F}' \setminus \mathcal{S}|+2}} \left( \sum_{\mathbf{y} \in \Upsilon(\mathcal{S}; \mathbf{v}_{1..m})} \kappa_3(\mathbf{y}) + \sum_{\mathbf{y} \in \Upsilon(\mathcal{S}; \mathbf{v}_{1..m})} \kappa_2(\mathbf{y}) \right)
$$

2. For the feature $n - 1$, consider an arbitrary subset $\mathcal{S} \cup \{n\}$, then:

$$
\phi(\mathcal{S} \cup \{n - 1, n\}; \mathcal{M}, \mathbf{v}_{1..n}) - \phi(\mathcal{S} \cup \{n\}; \mathcal{M}, \mathbf{v}_{1..n}) \tag{28}
$$

$$
= \left( \frac{1}{2^{|\mathcal{F}' \setminus \mathcal{S}|}} \sum_{\mathbf{x}' \in \Upsilon(\mathcal{S} \cup \{n-1,n\}; \mathbf{v}_{1..n})} \kappa(\mathbf{x}') \right) - \left( \frac{1}{2^{|\mathcal{F}' \setminus \mathcal{S}|+1}} \sum_{\mathbf{x}' \in \Upsilon(\mathcal{S} \cup \{n\}; \mathbf{v}_{1..n})} \kappa(\mathbf{x}') \right)
$$

$$
= \frac{1}{2^{|\mathcal{F}' \setminus \mathcal{S}|+1}} \left( \sum_{\mathbf{y} \in \Upsilon(\mathcal{S}; \mathbf{v}_{1..m})} (\kappa_1(\mathbf{y}) \vee \kappa_3(\mathbf{y})) - \sum_{\mathbf{y} \in \Upsilon(\mathcal{S}; \mathbf{v}_{1..m})} \kappa_1(\mathbf{y}) \right)
$$

$$
= \frac{1}{2^{|\mathcal{F}' \setminus \mathcal{S}|+1}} \left( \sum_{\mathbf{y} \in \Upsilon(\mathcal{S}; \mathbf{v}_{1..m})} \kappa_3(\mathbf{y}) \right)
$$

Thus, we can conclude that $\mathsf{Sv}(n - 1) > 0$. For any subset $\mathcal{S} \subseteq \mathcal{F}'$, we now focus on the feature $n$.

744   1. For the feature $n$, consider an arbitrary subset $\mathcal{S} \subseteq \mathcal{F}'$ and without the feature $n-1$, then:

$$\phi(\mathcal{S} \cup \{n\}; \mathcal{M}, \mathbf{v}_{1..n}) - \phi(\mathcal{S}; \mathcal{M}, \mathbf{v}_{1..n}) \tag{29}$$

$$= \left( \frac{1}{2^{|\mathcal{F}' \setminus \mathcal{S}| + 1}} \sum_{\mathbf{x}' \in \Upsilon(\mathcal{S} \cup \{n\}; \mathbf{v}_{1..n})} \kappa(\mathbf{x}') \right) - \left( \frac{1}{2^{|\mathcal{F}' \setminus \mathcal{S}| + 2}} \sum_{\mathbf{x}' \in \Upsilon(\mathcal{S}; \mathbf{v}_{1..n})} \kappa(\mathbf{x}') \right)$$

$$= \frac{1}{2^{|\mathcal{F}' \setminus \mathcal{S}| + 1}} \left( \sum_{\mathbf{y} \in \Upsilon(\mathcal{S}; \mathbf{v}_{1..m})} (\kappa_1(\mathbf{y}) \vee \kappa_3(\mathbf{y})) + \sum_{\mathbf{y} \in \Upsilon(\mathcal{S}; \mathbf{v}_{1..m})} \kappa_1(\mathbf{y}) \right)$$

$$- \frac{1}{2^{|\mathcal{F}' \setminus \mathcal{S}| + 2}} \left( \sum_{\mathbf{y} \in \Upsilon(\mathcal{S}; \mathbf{v}_{1..m})} (\kappa_1(\mathbf{y}) \vee \kappa_3(\mathbf{y})) + \sum_{\mathbf{y} \in \Upsilon(\mathcal{S}; \mathbf{v}_{1..m})} \kappa_1(\mathbf{y}) \right)$$

$$- \frac{1}{2^{|\mathcal{F}' \setminus \mathcal{S}| + 2}} \left( \sum_{\mathbf{y} \in \Upsilon(\mathcal{S}; \mathbf{v}_{1..m})} (\kappa_1(\mathbf{y}) \vee \kappa_2(\mathbf{y})) + \sum_{\mathbf{y} \in \Upsilon(\mathcal{S}; \mathbf{v}_{1..m})} \kappa_1(\mathbf{y}) \right)$$

$$= \frac{1}{2^{|\mathcal{F}' \setminus \mathcal{S}| + 2}} \left( \sum_{\mathbf{y} \in \Upsilon(\mathcal{S}; \mathbf{v}_{1..m})} (\kappa_1(\mathbf{y}) \vee \kappa_3(\mathbf{y})) - \sum_{\mathbf{y} \in \Upsilon(\mathcal{S}; \mathbf{v}_{1..m})} (\kappa_1(\mathbf{y}) \vee \kappa_2(\mathbf{y})) \right)$$

$$= \frac{1}{2^{|\mathcal{F}' \setminus \mathcal{S}| + 2}} \left( \sum_{\mathbf{y} \in \Upsilon(\mathcal{S}; \mathbf{v}_{1..m})} \kappa_3(\mathbf{y}) - \sum_{\mathbf{y} \in \Upsilon(\mathcal{S}; \mathbf{v}_{1..m})} \kappa_2(\mathbf{y}) \right)$$

745   2. For the feature $n$, consider an arbitrary subset $\mathcal{S} \cup \{n-1\}$, then:

$$\phi(\mathcal{S} \cup \{n-1, n\}; \mathcal{M}, \mathbf{v}_{1..n}) - \phi(\mathcal{S} \cup \{n-1\}; \mathcal{M}, \mathbf{v}_{1..n}) \tag{30}$$

$$= \left( \frac{1}{2^{|\mathcal{F}' \setminus \mathcal{S}|}} \sum_{\mathbf{x}' \in \Upsilon(\mathcal{S} \cup \{n-1, n\}; \mathbf{v}_{1..n})} \kappa(\mathbf{x}') \right) - \left( \frac{1}{2^{|\mathcal{F}' \setminus \mathcal{S}| + 1}} \sum_{\mathbf{x}' \in \Upsilon(\mathcal{S} \cup \{n-1\}; \mathbf{v}_{1..n})} \kappa(\mathbf{x}') \right)$$

$$= \frac{1}{2^{|\mathcal{F}' \setminus \mathcal{S}| + 1}} \left( \sum_{\mathbf{y} \in \Upsilon(\mathcal{S}; \mathbf{v}_{1..m})} (\kappa_1(\mathbf{y}) \vee \kappa_3(\mathbf{y})) - \sum_{\mathbf{y} \in \Upsilon(\mathcal{S}; \mathbf{v}_{1..m})} (\kappa_1(\mathbf{y}) \vee \kappa_2(\mathbf{y})) \right)$$

$$= \frac{1}{2^{|\mathcal{F}' \setminus \mathcal{S}| + 1}} \left( \sum_{\mathbf{y} \in \Upsilon(\mathcal{S}; \mathbf{v}_{1..m})} \kappa_3(\mathbf{y}) - \sum_{\mathbf{y} \in \Upsilon(\mathcal{S}; \mathbf{v}_{1..m})} \kappa_2(\mathbf{y}) \right)$$

746   Note that $\phi(\mathcal{S} \cup \{n\}; \mathcal{M}, \mathbf{v}_{1..n}) - \phi(\mathcal{S}; \mathcal{M}, \mathbf{v}_{1..n}) < 0$ and $\phi(\mathcal{S} \cup \{n-1, n\}; \mathcal{M}, \mathbf{v}_{1..n}) - \phi(\mathcal{S} \cup \{n-$
747   $1\}; \mathcal{M}, \mathbf{v}_{1..n}) < 0$ if and only if $\mathcal{S} = \mathcal{F}'$. For any other proper subset $\mathcal{S} \subset \mathcal{F}'$, $\phi(\mathcal{S} \cup \{n\}; \mathcal{M}, \mathbf{v}_{1..n}) -$
748   $\phi(\mathcal{S}; \mathcal{M}, \mathbf{v}_{1..n}) \geq 0$ and $\phi(\mathcal{S} \cup \{n-1, n\}; \mathcal{M}, \mathbf{v}_{1..n}) - \phi(\mathcal{S} \cup \{n-1\}; \mathcal{M}, \mathbf{v}_{1..n}) \geq 0$.

749   Moreover, for a fixed set $\mathcal{S} \subseteq \mathcal{F}'$, we have $(\phi(\mathcal{S} \cup \{n-1\}; \mathcal{M}, \mathbf{v}_{1..n}) - \phi(\mathcal{S}; \mathcal{M}, \mathbf{v}_{1..n})) > (\phi(\mathcal{S} \cup$
750   $\{n\}; \mathcal{M}, \mathbf{v}_{1..n}) - \phi(\mathcal{S}; \mathcal{M}, \mathbf{v}_{1..n}))$, and $(\phi(\mathcal{S} \cup \{n-1, n\}; \mathcal{M}, \mathbf{v}_{1..n}) - \phi(\mathcal{S} \cup \{n\}; \mathcal{M}, \mathbf{v}_{1..n})) >$
751   $(\phi(\mathcal{S} \cup \{n-1, n\}; \mathcal{M}, \mathbf{v}_{1..n}) - \phi(\mathcal{S} \cup \{n-1\}; \mathcal{M}, \mathbf{v}_{1..n}))$. Therefore, $\mathsf{Sv}(n-1) > \mathsf{Sv}(n)$.

In the following, we prove that $\mathsf{Sv}(n-1) + \mathsf{Sv}(n) > 0$ by focusing on all subsets $\mathcal{S} \subseteq \mathcal{F}$ where $m - 2 \leq |\mathcal{S}| \leq m + 1$. For the feature $n - 1$, we have:

$$\sum_{\substack{\mathcal{S} \subseteq \mathcal{F} \setminus \{n-1\} \\ m-2 \leq |\mathcal{S}| \leq m+1}} \frac{|\mathcal{S}|!(m+2-|\mathcal{S}|-1)!}{(m+2)!} \times (\phi(\mathcal{S} \cup \{n-1\}; \mathcal{M}, \mathbf{v}_{1..n}) - \phi(\mathcal{S}; \mathcal{M}, \mathbf{v}_{1..n})) \quad (31)$$

$$= \sum_{|\mathcal{S}|=m+1, n \in \mathcal{S}} \frac{(m+1)!(m+1-(m+1))!}{(m+2)!} \times \frac{1}{2^{m-m+1}} \times 0$$

$$+ \sum_{|\mathcal{S}|=m, n \in \mathcal{S}} \frac{m!(m+1-m)!}{(m+2)!} \times \frac{1}{2^{m-(m-1)+1}} \times 1$$

$$+ \sum_{|\mathcal{S}|=m, n \notin \mathcal{S}} \frac{m!(m+1-m)!}{(m+2)!} \times \frac{1}{2^{m-m+2}} \times (0+1)$$

$$+ \sum_{|\mathcal{S}|=m-1, n \in \mathcal{S}} \frac{(m-1)!(m+1-(m-1))!}{(m+2)!} \times \frac{1}{2^{m-(m-2)+1}} \times 2$$

$$+ \sum_{|\mathcal{S}|=m-1, n \notin \mathcal{S}} \frac{(m-1)!(m+1-(m-1))!}{(m+2)!} \times \frac{1}{2^{m-(m-1)+2}} \times (1+1)$$

$$+ \sum_{|\mathcal{S}|=m-2, n \notin \mathcal{S}} \frac{(m-2)!(m+1-(m-2))!}{(m+2)!} \times \frac{1}{2^{m-(m-2)+2}} \times (2+1)$$

$$= \frac{8m+13}{16(m+2)(m+1)}$$

For the feature $n$, we have:

$$\sum_{\substack{\mathcal{S} \subseteq \mathcal{F} \setminus \{n\} \\ m-2 \leq |\mathcal{S}| \leq m+1}} \frac{|\mathcal{S}|!(m+2-|\mathcal{S}|-1)!}{(m+2)!} \times (\phi(\mathcal{S} \cup \{n\}; \mathcal{M}, \mathbf{v}_{1..n}) - \phi(\mathcal{S}; \mathcal{M}, \mathbf{v}_{1..n})) \quad (32)$$

$$= \sum_{|\mathcal{S}|=m+1, n-1 \in \mathcal{S}} \frac{(m+1)!(m+1-(m+1))!}{(m+2)!} \times \frac{1}{2^{m-m+1}} \times (0-1)$$

$$+ \sum_{|\mathcal{S}|=m, n-1 \in \mathcal{S}} \frac{m!(m+1-m)!}{(m+2)!} \times \frac{1}{2^{m-(m-1)+1}} \times (1-1)$$

$$+ \sum_{|\mathcal{S}|=m, n-1 \notin \mathcal{S}} \frac{m!(m+1-m)!}{(m+2)!} \times \frac{1}{2^{m-m+2}} \times (0-1)$$

$$+ \sum_{|\mathcal{S}|=m-1, n-1 \in \mathcal{S}} \frac{(m-1)!(m+1-(m-1))!}{(m+2)!} \times \frac{1}{2^{m-(m-2)+1}} \times (2-1)$$

$$+ \sum_{|\mathcal{S}|=m-1, n-1 \notin \mathcal{S}} \frac{(m-1)!(m+1-(m-1))!}{(m+2)!} \times \frac{1}{2^{m-(m-1)+2}} \times (1-1)$$

$$+ \sum_{|\mathcal{S}|=m-2, n-1 \notin \mathcal{S}} \frac{(m-2)!(m+1-(m-2))!}{(m+2)!} \times \frac{1}{2^{m-(m-2)+2}} \times (2-1)$$

$$= \frac{-6m-11}{16(m+2)(m+1)}$$

Their summation is $\frac{m+1}{8(m+2)(m+1)}$, since $m \geq 2$, $\mathsf{Sv}(n-1) + \mathsf{Sv}(n) > 0$. Thus, it can be concluded that for the irrelevant feature $n - 1$ and the relevant feature $n$, $|\mathsf{Sv}(n-1)| > |\mathsf{Sv}(n)|$. $\qquad\square$

**Corollary 1.** For any $n \geq 7$, there exist boolean functions defined on $n$ variables, and at least one instance, for which there exists an irrelevant feature $i_1 \in \mathcal{F}$, and a relevant feature $i_2 \in \mathcal{F} \setminus \{i_1\}$, such that $\mathsf{Sv}(i_1) > \mathsf{Sv}(i_2) > 0$.

 *Proof.* We utilize the function constructed in Proposition 7, which is given by:

$$\kappa(\mathbf{x}_{1..m}, x_{n-1}, x_n) := \begin{cases} \kappa_1(\mathbf{x}_{1..m}) & \text{if } x_{n-1} = 0 \\ \kappa_1(\mathbf{x}_{1..m}) \vee \kappa_2(\mathbf{x}_{1..m}) & \text{if } x_{n-1} = 1 \wedge x_n = 0 \\ \kappa_1(\mathbf{x}_{1..m}) \vee \kappa_3(\mathbf{x}_{1..m}) & \text{if } x_{n-1} = 1 \wedge x_n = 1 \end{cases} \qquad (33)$$

However, we choose a different function $\kappa_3$ that satisfies the following condition: for any point $\mathbf{x}_{1..m}$ such that $d_H(\mathbf{x}_{1..m}, \mathbf{v}_{1..m}) \leq 2$, where $d_H(\cdot)$ represents the Hamming distance, $\kappa_3(\mathbf{x}_{1..m}) = 1$. According to the proof of Proposition 7, it can be derived that $\mathsf{Sv}(n-1) > 0$ and $\mathsf{Sv}(n-1) > \mathsf{Sv}(n)$. In the following, we prove that $\mathsf{Sv}(n) > 0$ by focusing on all subsets $\mathcal{S} \subseteq \mathcal{F} \setminus \{n\}$ where $m - 4 \leq |\mathcal{S}| \leq m + 1$, and show that the sum of their values is greater than 0, which implies that $\mathsf{Sv}(n) > 0$ when considering all possible subsets $\mathcal{S}$.

$$\sum_{\substack{\mathcal{S} \subseteq \mathcal{F} \setminus \{n\} \\ m-4 \leq |\mathcal{S}| \leq m+1}} \frac{|\mathcal{S}|!(m + 2 - |\mathcal{S}| - 1)!}{(m+2)!} \times (\phi(\mathcal{S} \cup \{n\}; \mathcal{M}, \mathbf{v}_{1..n}) - \phi(\mathcal{S}; \mathcal{M}, \mathbf{v}_{1..n})) \qquad (34)$$

$$= \sum_{|\mathcal{S}|=m+1, n-1 \in \mathcal{S}} \frac{(m+1)!(m + 1 - (m+1))!}{(m+2)!} \times \frac{1}{2^{m-m+1}} \times (0 - 1)$$

$$+ \sum_{|\mathcal{S}|=m, n-1 \in \mathcal{S}} \frac{m!(m + 1 - m)!}{(m+2)!} \times \frac{1}{2^{m-(m-1)+1}} \times (1 - 1)$$

$$+ \sum_{|\mathcal{S}|=m, n-1 \notin \mathcal{S}} \frac{m!(m + 1 - m)!}{(m+2)!} \times \frac{1}{2^{m-m+2}} \times (0 - 1)$$

$$+ \sum_{|\mathcal{S}|=m-1, n-1 \in \mathcal{S}} \frac{(m-1)!(m + 1 - (m-1))!}{(m+2)!} \times \frac{1}{2^{m-(m-2)+1}} \times \left( \binom{2}{1} + \binom{2}{2} - 1 \right)$$

$$+ \sum_{|\mathcal{S}|=m-1, n-1 \notin \mathcal{S}} \frac{(m-1)!(m + 1 - (m-1))!}{(m+2)!} \times \frac{1}{2^{m-(m-1)+2}} \times (1 - 1)$$

$$+ \sum_{|\mathcal{S}|=m-2, n-1 \in \mathcal{S}} \frac{(m-2)!(m + 1 - (m-2))!}{(m+2)!} \times \frac{1}{2^{m-(m-3)+1}} \times \left( \binom{3}{1} + \binom{3}{2} - 1 \right)$$

$$+ \sum_{|\mathcal{S}|=m-2, n-1 \notin \mathcal{S}} \frac{(m-2)!(m + 1 - (m-2))!}{(m+2)!} \times \frac{1}{2^{m-(m-2)+2}} \times \left( \binom{2}{1} + \binom{2}{2} - 1 \right)$$

$$+ \sum_{|\mathcal{S}|=m-3, n-1 \in \mathcal{S}} \frac{(m-3)!(m + 1 - (m-3))!}{(m+2)!} \times \frac{1}{2^{m-(m-4)+1}} \times \left( \binom{4}{1} + \binom{4}{2} - 1 \right)$$

$$+ \sum_{|\mathcal{S}|=m-3, n-1 \notin \mathcal{S}} \frac{(m-3)!(m + 1 - (m-3))!}{(m+2)!} \times \frac{1}{2^{m-(m-3)+2}} \times \left( \binom{3}{1} + \binom{3}{2} - 1 \right)$$

$$= \frac{11m - 47}{32(m+2)(m+1)}$$

Since $m \geq 5$, for all subsets $\mathcal{S} \subseteq \mathcal{F} \setminus \{n\}$ where $m - 4 \leq |\mathcal{S}| \leq m + 1$, their summation is greater than 0. This implies that $\mathsf{Sv}(n) > 0$. Thus, it can be concluded that for the irrelevant feature $n - 1$ and the relevant feature $n$, we have $\mathsf{Sv}(n-1) > \mathsf{Sv}(n) > 0$. □

**Proposition 8.** For Propositions 3 to 5, and Proposition 7 the following are lower bounds on the numbers issues exhibiting the respective issues:

1. For Proposition 3, a lower bound on the number of functions exhibiting I1 is $2^{2^{(n-1)}} - n - 3$.

2. For Proposition 4, a lower bound on the number of functions exhibiting I3 is $2^{2^{(n-1)/2}} - 2$.

3. For Proposition 5, a lower bound on the number of functions exhibiting I4 is $2^{2^{(n-2)/2}} - 2$.

4. For Proposition 7, a lower bound on the number of functions exhibiting I2 is $2^{2^{n-2} - (n-2) - 1} - 1$.

*Sketch.* For Proposition 3, there exist $2^{2^{(n-1)}} - n - 3$ distinct non-constant functions $\kappa_2$. For each such function $\kappa_2$, $\kappa_1$ can be defined by changing the prediction of some points predicted as 1 by $\kappa_2$ to 0. It is evident that $\kappa_1 \models \kappa_2$ but $\kappa_1 \neq \kappa_2$.

779 For Propositions 4 and 5, there exist $2^{2^{(n-1)/2}} - 2$ distinct non-constant functions $\kappa_1$. We can then
780 define $\kappa_2$ by renaming each variable $x_i$ of $\kappa_1$ with a new variable $x_{m+i}$.

781 For Proposition 7, the functions $\kappa_2$ and $\kappa_3$ are assumed to be fixed, while the flexibility lies in
782 the choice of $\kappa_1$ ($\kappa_1$ can be 0 but cannot be 1). As $\kappa_2$ covers 1 point and $\kappa_3$ covers $n - 2$ points,
783 the remaining points in the feature space can be used to define the function $\kappa_1$. Thus, there are
784 $2^{2^{n-2}-(n-2)-1} - 1$ possible functions for $\kappa_1$. □