# OpenReview forum: "A Refutation of Shapley Values for Explainability"
_NeurIPS.cc/2023/Conference — Submitted to NeurIPS 2023_

### Official Review · Reviewer_QRCo · 2023-06-26

**Soundness:** 3 good
**Presentation:** 4 excellent
**Contribution:** 2 fair
**Rating:** 4
**Confidence:** 4

**Summary:**

In this paper, the authors formally define five anomalies for an
explainability score and prove that for every n >= 4, there exist
Boolean classifiers defined over n features that exhibit one or more
of these anomalies for the SHAP score. In this way, the authors
provide evidence of the inadequacy of Shapley values for
explainability.

The aforementioned anomalies are defined by considering the concept of
abductive explanation. More precisely, given a binary classification
model M : {0,1}^n -> {0,1} and a tuple v in {0,1}^n, a subset X of {1,
..., n} is said to be a weak abductive explanation of (M,v) if for
every y in {0,1}^n such that y[i] = v[i] for every i in X, it holds
that M(y) = M(v). In other words, the values of v for the features in
X are enough to obtain the same result as M(v), so they are enough to
explain the output of M for v. Moreover, a subset X of {1, ..., n} is
said to be an abductive explanation of (M,v) if X is a weak abductive
explanation of (M,v), and there is no weak abductive explanation X' of
(M,v) such that X' is a proper subset of X. In other words, X is an
abductive explanation for (M,v) if X is a minimal weak abductive
explanation for (M,v). Then a feature i is said to be relevant for
(M,v) if there exists an abductive explanation X of (M,v) such that i
belongs to X, and otherwise i is said to be irrelevant for (M,v). With
this notion of irrelevance, the anomaly I5 for the SHAP score is
defined as the existence of a feature i such that i is irrelevant for
(M,v), but the absolute value of the SHAP score of i is greater than
the absolute value of the SHAP score of every other feature. Thus,
this can be considered as an anomaly of the SHAP score, as i is an
irrelevant feature that is considered more relevant according to the
SHAP score that all the other features (some of which are
relevant). The other four anomalies considered in the paper (I1, I2,
I3, I4) are defined in a similar fashion.

**Strengths:**

1. The five notions of anomaly studied in the paper clearly represent
anomalies for explainability scores. These notions are properly
formalized in the paper.

2. The paper provides valuable insights into the SHAP score,
specifically providing a formal framework to assess its adequacy as an
explainability score.

3. The paper provides one of the first formal results of the
inadequacy of Shapley values for explainability.

4. The paper is well written.

**Weaknesses:**

1. The results of the paper show that a tiny proportion of the Boolean
classifiers defined over n features exhibit some of the anomalies I1,
I2, I3, I4 or I5. For example, the paper proves that at least
2^{2^{n-1} - n - 3} Boolean classifiers exhibits anomaly I1, which is
a tiny proportion of the 2^{2^n} possible Boolean classifiers defined
over n features. Hence, it could be the case that the vast majority of
Boolean classifiers do not exhibit the anomalies studied in the paper.

2. In practice Boolean classifiers are given in some specific
formalism, such as decision trees or binary decision diagrams. The
authors do not provide any results about the formalisms that are
suitable to express the Boolean functions exhibiting anomalies. For
example, is it possible to express the Boolean functions in the proofs
of Propositions 3, 4, 5 and 6 as decision trees of polynomial size in
the number n of features? If this is not possible, can these functions
be expressed as FBDDs (or d-DNNFs) of polynomial size in the number n
of features?

**Questions:**

Could you please comment on the point 1. and 2. mentioned in
Weaknesses.


**Limitations:**

The following are the main limitations of this work (see Weaknesses),
which are not addressed in the paper.

- The results of the paper show that a tiny proportion of the Boolean
classifiers defined over n features exhibits some of the anomalies I1,
I2, I3, I4 or I5.

- The authors do not provide any results about the practical
formalisms (such as decision trees) that are suitable to express the
Boolean functions exhibiting anomalies.

---

> ### Author Rebuttal · Authors · 2023-08-09
>
> There is a misunderstanding in the review.
>
> The bounds proved in the paper are *lower* bounds, and it is stated in
> the paper that these are fairly loose lower bounds. The goal of these
> bounds is solely to establish that the number of boolean classifiers
> for which Shapley values exhibit some sort of issue is non-negligible.
>
> Q1. We dispute the claim made by the reviewer that the number of
> boolean classifiers is "tiny". First, this is not the case because the
> paper only aims at proposing lower bounds on the numbers of such
> classifiers. Also, and as stated in the paper, the lower bounds do not
> aim to be tight. Quoting from our paper: "we can prove the following
> (fairly loose) lower bounds on the number of functions exhibiting the
> different issues". Second, the number is not "tiny" because, as
> demonstrated by the experimental results reported in reference [35],
> for some of the issues with Shapley values, almost *all* boolean
> classifiers exhibit those issues. Furthermore, even if the number of
> boolean classifiers were indeed "tiny", the results in our paper prove
> that one of the most widely used explainability methods can produce
> misleading information regarding relative feature importance, for
> arbitrarily many boolean classifiers. Even if the number of such
> classifiers was indeed negligible (and it is not), the fact that the
> theoretical foundation of several explainability methods is flawed is
> reason for serious concern.
>
> Q2: This is an interesting question, but one that it orthogonal to the
> goals of the paper. The paper proves that there are arbitrarily many
> boolean classifiers for which Shapley values will give misleading
> information regarding relative feature importance. Also, given the
> experimental results in [35], it is guaranteed that there exist many
> boolean classifiers, many of which are easy to represent either with
> decision trees or with tractable circuits, and for which the size of
> the representation is polynomial on the number of features. We can add
> this comment to the paper, but it does not affect the paper's claims
> in any way.

---

> > ### Comment · Reviewer_QRCo · 2023-08-15
> > **Response to rebuttal**
> >
> > Thank you for your answer.
> >
> > I understand that the bounds provided in the paper represent lower bounds and are not tight. My point is that the bounds proved in the paper show that a tiny proportion of Boolean classifiers, defined over n features, exhibit some anomalies. More formally, if f(n) is this fraction as a function of n, then lim_{n -> infinity} f(n) = 0. Obviously, this does not preclude the existence of stronger lower bounds that demonstrate that a non-negligible proportion of Boolean classifiers exhibit some anomalies.
> >
> > While I believe this paper offers valuable insights, I still think it is not ready for publication. It does not show either that a significant proportion of Boolean classifiers exhibit the anomalies discussed in the paper or that popular formalisms for Boolean classifiers display such anomalies.

---

> > > ### Author Response · Authors · 2023-08-17
> > >
> > > We thank the reviewer for the comments. However, we disagree with some of the comments.
> > >
> > > First, and as stated in our rebuttal, it is already known that, for boolean classifiers with 4 features, some of the issues studied in our paper occur in almost *all* classifiers. So, it is already known that the issues studied in our paper occur in a *large* fraction (in fact in *most$) boolean classifiers. To be clear, issue I1 is identified in 99.67% of all boolean classifiers with 4 features. Issue I2 is identified in 61.72% of all boolean classifiers with 4 features. These numbers clearly indicate that the issues reported in the paper are most often observed.
> > >
> > > We can include an extended table with these results, which prove that the issues studied in our paper occur in most of the boolean classifiers one can think about.
> > >
> > > Second, and also as stated in our rebuttal, for some of the classifiers that exhibit one or more of the reported issues, their representation is polynomial on the number of features. This is really not an issue.
> > >
> > > Third, and as stated in a comment to another reviewer, let us agree that to disprove a theory a single counterexample suffices. Earlier work revealed the existence of issues for a large fraction of boolean classifiers with four features. This result might be challenged because of the *fixed* number of features. Our paper proves that the issues reported in earlier work, and also a number of additional issues, occur in arbitrary many boolean classifiers. This disproves the existing theory on Shapley values for XAI, independently of how frequently those issues might occur (and existing results prove that they occur in almost all boolean classifiers with four features).
> > >
> > > We will be happy to provide additional clarifications, but we feel that the two criticisms raised by the reviewer have been sufficiently deconstructed.

---

### Official Review · Reviewer_4AZv · 2023-07-06

**Soundness:** 3 good
**Presentation:** 3 good
**Contribution:** 2 fair
**Rating:** 4
**Confidence:** 4

**Summary:**

This paper reviews previous work on ideas of feature importance and hi-lighted inconsistencies with Shaley values. It defines ideas of importance and irrelevance of features in a Boolean ML model. These definitions are based on the idea of a minimal set of inputs needed to freeze an model output. necessary inputs are in every minimal coalition that can freeze the output, relevant inputs are in at least one minimal coalition, and irrelevant inputs are in no coalitions. They then go on to show that, among other issues, there exist Boolean models and certain inputs where irrelevant inputs are given large Shapley values, while relevant inputs are given a Shapley value of zero. Thus, the logic goes, Shapley values do not track importance.

The paper's original contributions are to prove that model/input pairs with issues exist/can be found for models of any input size. Previously, only small models were exhibited to have these issues, but it was unknown if larger models also had these issues. They also give lower bounds on the number of models that have these issues.

**Strengths:**

- Generally clear and straightforward exposition.
- Good background and presentation of previous results.
- Results are easy to understand.
- idea of necessary, relevant, and irrelevant is intuitive.



**Weaknesses:**

- Paper is based on a comparison of apples to oranges, without an in-depth analysis of the issue. It is possible that the whole paper is based on a misunderstanding. Further analysis is needed.
- Some grammatical issues.
- Contributions are not very significant.

**Questions:**

- I believe, based on my calculations, that the paper's definition of Shapley Values in equation (3) is incorrect. For example, take model $F(x_1, x_2) = x_1$. I get that the Shapely values of the input $(1,1)$, assuming the baseline is $(0,0)$, is 1/2 for $x_1$ and 0 for $x_2$ for your definition. The Shapley value is actually 1 for $x_1$ and 0 for $x_2$, based on widely known definitions. The Shapley value satisfies the axiom of completeness, or efficiency, so the definition you gave is definitely not correct if it indeed gives the values 1/2 and 0.
- The heart of the paper is a critique of Shapley values. It is asserted that there is a metric of feature importance and irrelevance and that the Shapley value assigns importance to what this other metric indicates is not important, and assigns no importance to what this other metric identifies as important. Admittedly this critique is from another paper, and this paper builds on this critique.

Fundamentally, the Shaley Value indicates a feature's contribution to function value change in comparison to the function evaluated at the comparative baseline input. The metrics of AXp and CXp are metrics of the ability of inputs to determine or alter a model at an input. One is about function change from a baseline, the other is about fixing or altering an output. There does not seem to be any analysis as to whether these notions are tracking the same underlying, and undefined, concept of "importance." This paper defined "important" one way, while Shaley values define it another way. Is the problem that Shapley values do not indicate importance, or is the problem that Shapley values indicate importance according to one definition, while people are confused and think it indicates importance by another definition? The second case would account for this papers results, while problematizing the conclusion that the Shapley values do not indicate importance at all. This question needs further investigation and exposition in the paper, I believe.

As an illustration, let $F$ be a model with one input defined as $F(x) = x$. Also, let the input of consideration be $x=0$. By the definition of the paper, $x$ very important because $x$ is a "necessary" input. However, The Shaley value of $x$ is $0$ at the input $0$. An inadequate analysis of this results it that the Shapley value indicated zero importance to a "necessary" input, so the Shaley value does not track importance. An equally inadequate analysis, opposite of the first, is that a "necessary" input did not cause any function change, so the idea of a "necessary" input is flawed. However, a more sophisticated analysis is that necessary inputs can have no contribution to a model changing from the baseline. Shapley values measure one thing, and "necessary" inputs measure another.

In summary, it appears possible that this paper deals with two different metrics of "important," that these metrics disagree, but also, the two metrics measure different things and work in different ways. Shapley values do not indicate importance according to this paper's definition, but why is that necessarily an issue with Shapley values?


Minor issues:
-  (39) "contributions of features to explainability". Do you mean function output?
- (97)  $2^\mathcal{F}$ should be $2^{|\mathcal{F}|}$ ? Sometimes you use one, sometimes another.
- (108) "did" contribute", not "can" contribute
- (117-118) "which corresponds to a PI-explanations" odd grammar.
- please define A PI-explanations, prime implicants. Seems that unknowns are defined in terms of unknowns.
- Unsure if WAXp is a function?
- I1-I5 are stated in what appears to be sentence fragments.
- I5 -> I2, but also, I5-> I1, and I4 -> I3, I1, and I2.

**Limitations:**

The author has not discussed the limitations of the claim that Shapley values are refuted. This statement seems not entirely supported. See questions.

---

> ### Author Rebuttal · Authors · 2023-08-09
>
> We thank the reviewer for the in-depth review. We feel there is a
> misunderstanding in what is being proved.
>
> Our work builds on the definition of Shapley values for XAI studied in
> recent work, namely references [7,8,21,22], but also the more recently
> published paper:
>
> [78] M. Arenas, P. Barceló, L. E. Bertossi, M. Monet: On the
> Complexity of SHAP-Score-Based Explanations: Tractability via
> Knowledge Compilation and Non-Approximability Results.
> J. Mach. Learn. Res. 24: 63:1-63:58 (2023)
>
> These papers are based on the NeurIPS'17 paper by Lundberg&Lee
> (reference [47] in our paper), which builds on earlier work on the
> same topic. The definition of Shapley values for XAI used in our paper
> is taken verbatim from those papers, specifically [7,8,78] and
> indirectly [21,22].
>
> Furthermore, we underscore that existing bibliography concurs with our
> interpretation of Shapley values (for XAI) as a measure of feature
> importance, and the meaning of 'importance'. Concretely, references
> [8,78] read: "Thus, SHAP(M,e,x) is a weighted average of the
> contribution of feature x on e to the classification result, ...".
> Furthermore, references [21,22] read: "Finally, the SHAP explanation
> computes a score for each feature $X\in\mathbf{X}$ averaged over all
> possible contexts, and thus measures the influence feature X has on
> the outcome." More importantly, our paper also includes quotes from
> references [64] and [65] further supporting this interpretation of
> Shapley values (for XAI). To be clear: the interpretation of Shapley
> value as a measure of feature importance in those papers and in our
> paper is exactly the same. Thus, the comparison is not 'apples to
> oranges'; quite the contrary. We ask the review to be changed
> accordingly.
>
> We also underscore that relevancy has been studied in logic-based
> abduction studied since the 90s, e.g. reference [23] in our
> paper. As stated in [23] (and adapting to XAI), relevant features
> occur in some acceptable explanation; whereas irrelevant features do
> not occur in any. AXp's are concerned with minimal conditions for
> prediction sufficiency; irrelevant features do not occur in any AXp.
> CXp's are concerned with minimal conditions for prediction change;
> irrelevant features do not occur in any CXp. The key point here is
> that irrelevant features play no role whatsoever, neither in
> prediction sufficiency, nor in prediction change. Thus, assigning no
> importance to relevant features is misleading; and assigning
> importance to irrelevant features is also misleading.
>
> The reviewer makes the valid point that there exist other
> interpretations of Shapley values for XAI where the set function is
> defined differently, and where baselines are considered. Besides the
> NeurIPS'17 paper, where the concept of base value is described, there
> are other works formalized the use of baselines, namely:
>
> D. Janzing, L. Minorics, P. Blöbaum: Feature relevance quantification
> in explainable AI: A causal problem. AISTATS 2020: 2907-2916
>
> M. Sundararajan, A. Najmi: The Many Shapley Values for Model
> Explanation. ICML 2020: 9269-9278
>
> However, considering different baselines is not the focus of our work,
> as it would be close to impossible to refute all the different
> heuristics that have been proposed when approximating Shapley values
> for XAI. Our paper focuses on a simple, yet rigorous, definition of
> Shapley values for explainability, as clarified above. Furthermore,
> our paper reveals the limitations of such a definition. We claim that
> this theoretical framework suffices, because the goal is to prove a
> counterexample to validity, and no sound theory withstands a single
> counterexample to validity. Nevertheless, in the updated version of
> the paper, we will include a statement to that effect:
>
> "The paper proves that a widely used definition of Shapley values for
> XAI can produce misleading information regarding relative feature
> importance. Nevertheless, it is left open whether other definitions of
> Shapley values, e.g. those based on considering different baselines
> [Refs], might circumvent the issues with Shapley values reported in
> this paper."
>
> Given the above, there is nothing incorrect in our paper. Our
> definition of Shapley values (for XAI) follows verbatim the
> definitions in earlier work (see [7,8,21,22]), including equation
> (3). The values computed with equation (3) are correct, given the
> definitions in our and in the earlier papers [7,8,21,22].
>
> Finally, we disagree with the reviewer regarding the comment:
> 'However, a more sophisticated analysis is that necessary inputs can
> have no contribution to a model changing from the baseline. Shapley
> values measure one thing, and "necessary" inputs measure another.'
> As with the rest of this review, the comment considers a concrete
> definition of the set function for Shapley values (for XAI) which is
> not the one our work is based on, and which is not the one used in the
> references cited in our paper. For the example given, the only way for
> a model to change the predicted value is to change the value of x.
> The Shapley value that we obtain in this case is -0.5 and the feature
> is relevant because it occurs in some AXp/CXp. So, all this makes
> complete sense. As stated earlier, coonsidering different baselines is
> not the focus of our paper, given the works our paper builds upon.

---

> > ### Comment · Reviewer_4AZv · 2023-08-17
> > **Response to Rebuttal**
> >
> > We thank the author for their clarification of the Shapley value. We agree that the formulation of Shapley values is in accord with previous literature, i.e. Lundberg and Lee (2017), and withdraw comments about the incorrectness of equation 3.
> >
> > Regarding the refutation of the "apples-to-oranges comparison" claim, we are not convinced. We agree that the paper, if correct, shows that for arbitrarily large domains, Shapley Values may not indicate necessary or irrelevant features. We disagree that this is a refutation of Shapley values for ML explainability.
> >
> > 1) It may be that Shapley values both ARE useful and legitimate for ML explainability AND do not track necessary or irrelevant features. This is because the purpose of Shapley values is to indicate feature contribution to function change relative to an input baseline. This purpose may be separate from necessary and irrelevant features.
> >
> > 2) Regarding the provided quotes: the word "importance" is not in any quote, only feature "contribution" and "influence." These quotes claim that Shapley values track feature contribution and influence, which different and more specific than feature importance. It is our opinion that the idea of "importance" is vague, and that when we take a high-resolution view of the matter, the issues seem to dissolve.
> >
> > 3) We conceded that some popular opinions may hold that Shapley values indicate feature importance, which is false for certain definition of importance. We disagree that the proper remedy is to assert Shapley values are not at all useful for ML explainability. We rather advocate for a clarification of what Shapley values are, and what they are not.
> >
> > We recommend that the paper move away from the "refutation" claim, and instead state the claim as: Shapley values are "incompatible" with a certain notion of feature importance.

---

> > > ### Author Response · Authors · 2023-08-17
> > >
> > > We thank the reviewer for the thoughtful comment, and for acknowledging that there is nothing wrong with our paper.
> > >
> > > However, we disagree with the reviewer in some of the comments made.
> > >
> > > 1. The reviewer states: "This is because the purpose of Shapley values is to indicate feature contribution to function change relative to an input baseline". As clarified in our rebuttal, we consider an existing definition of Shapley values for XAI with does *not* consider an input baseline. Furthermore, the papers upon which our work is based also do not consider an input baseline. Also, as stated in our rebuttal, our interpretation of the meaning of Shapley values for XAI is *exactly* the one used in those papers.
> > >
> > > 2. As already stated in our rebuttal, we will explicitly acknowledge that our refutation of Shapley values for XAI applies to an existing definition and interpretation of feature importance. However, for that definition and interpretation, what our paper establishes is indeed a refutation of Shapley values for XAI. Future work, ours or by others, will analyze the alternative definition of Shapley values that considers a baseline, given the now known result that, for some definitions of Shapley values for XAI, the obtained measures of feature importance are provably misleading.
> > >
> > > We will be happy to provide any additional clarifications, but we feel that the main criticisms raised by the reviewer have been been addressed.

---

### Official Review · Reviewer_6JiF · 2023-07-08

**Soundness:** 3 good
**Presentation:** 2 fair
**Contribution:** 1 poor
**Rating:** 3
**Confidence:** 3

**Summary:**

The paper demonstrates / constructs functions with features whose Shapley values (i.e., attributive importance in a prediction) is misaligned with their true relevance.

**Strengths:**

- Addresses a theoretical gap in our understanding of Shapley values.

**Weaknesses:**

- I find the problem being investigated to be mostly a mathematical curiosity that so happened to be open and has now been addressed.

**Questions:**

n/a

---

> ### Author Rebuttal · Authors · 2023-08-09
>
> This "review" is unacceptable in any credible conference.
>
> We fail to see how this "review" can even be accepted as a review.
> There are no concrete comments on the submitted work.
>
> The only stated strength reads: "Addresses a theoretical gap in our
> understanding of Shapley values." This is not true. Our paper does not
> address a theoretical gap in the understanding of Shapley values. It
> proves that Shapley values can produce misleading information regarding
> relative feature importance, for arbitrary many classifiers. Our paper
> does not address any 'gap'.
>
> Furthermore, the only listed weakness is a baseless statement about
> some 'mathematical curiosity', which has nothing to do with the focus
> of our paper. A paper that proves that, what thousands of earlier
> papers have used the theoretical justification for explainability, can
> produce misleading information is not a 'mathematical curiosity'.
>
> The lack of quality of this "review' has been reported to the Area
> Chairs, Senior Area Chairs and PC Chairs.

---

> > ### Comment · Reviewer_6JiF · 2023-08-10
> > **Response to Author Rebuttal**
> >
> > I acknowledge having read the response of the authors. I will try to elaborate more on my two points:
> >
> > 1. A "theoretical gap" is a missing piece in our understanding / knowledge of something. Prior to this work, we had a gap in our understanding / knowledge on whether "Shapley values can produce misleading information regarding relative feature importance, for arbitrary many classifiers". We didn't know whether this statement was true or not. Now, because of this paper, we know. The paper has closed this gap in our understanding / knowledge. I am not sure why the authors take issue with the claim that their work addresses a theoretical gap. How can this claim be possibly construed in a negative way?
> >
> > 2. The rebuttal claims that the paper proves "what thousands of earlier papers have used the theoretical justification for explainability, can produce misleading information". We already knew that. The paper already acknowledges that much in the first sentence of its abstract: "Recent work demonstrated the existence of Boolean functions for which Shapley values provide misleading information about the relative importance of features in rule-based explanations.". So, we knew that Shapley values "can produce misleading information". Much more modestly than what the rebuttal claims, this paper shows that Shapley values can not only produce misleading information as known, but can do so for arbitrarily many functions. I found this particular result to be of little consequence. The functions are artificially constructed, and the result, therefore, has no apparent implication on how widespread the problem with Shapley values (that we already knew was there) really is in practice. In my opinion, which I was asked to offer as a reviewer, the result of the paper is a mathematical curiosity: can we extend the existence result of previous work, to an "arbitrarily-many cases" result? The answer seems to be positive, without any obvious real-life repercussions.
> >
> > I stand to be corrected if the arbitrarily many functions considered in this paper somehow relate to functions used in the relevant literature.
> >
> > P.S. I thank the authors for being forthcoming in terms of reporting this review.

---

> > > ### Author Response · Authors · 2023-08-11
> > >
> > > Fact: the reviewer wrote two sentences in his/her "review".
> > > Given the reviewer's comment, we conclude that there was a lot the
> > > authors had to infer from those two lines. That is not how reviews are
> > > written, again not in any credible conference.
> > >
> > > Also, it should be said that some of the reviewer's comments are
> > > similar to the criticisms made by the other reviewers. We take this
> > > as a coincidence; but those comments should have been included when
> > > the review was written, not as an afterthought.
> > >
> > > What the reviewer claims that we (perhaps everybody working in XAI?)
> > > seem(s) to "know" comes from an arxiv preprint, never published. We
> > > emphasize: that preprint has not been published. Given the thousands
> > > of papers already published on Shapley values, and the hundreds that
> > > continue to being published almost every week, and given the ongoing
> > > proposed high-stakes uses of Shapley values, it seems evident that an
> > > arxiv preprint will not suffice to make sure that everybody
> > > understands what everybody seems to "know".
> > >
> > > The comments made by the reviewer, which were sadly not included in
> > > the review, merit a reply.
> > >
> > > 1. To close a "theoretical gap", one has to start from a sound theory.
> > > What reference [35] suggests, and this paper effectively demonstrates,
> > > is that Shapley values for XAI are unsound. So, there is no theory to
> > > start with, and so there is no gap to close.
> > >
> > > 2. As clarified in our paper, the fact that issues with Shapley values
> > > for XAI occur for boolean classifiers with four features could
> > > represent some sort of special case. For example, one might try to
> > > detect those special cases and then claim to have a sound theory. This
> > > paper proves otherwise, in that the number of special cases is
> > > unbounded. Evidently, that is why the result matters. All this is
> > > stated in our paper.
> > >
> > > 3. To disprove a theory, a single counterexample suffices. In the case
> > > of Shapley values for XAI, one might circumvent a few special cases.
> > > Therefore, the goal of our paper is to prove that that cannot be done,
> > > and so the theory of Shapley values for XAI is indeed unsound.
> > >
> > > The experimental results from [35], which the reviewer claims to be
> > > "known", show that the number of boolean classifiers exhibiting issues
> > > is actually massive, for classifiers with four features. So, asking
> > > "if the arbitrarily many functions considered in this paper somehow
> > > relate to functions used in the relevant literature" is really a moot
> > > point. Because of our paper, the use of Shapley values for XAI has now
> > > been disproved as a general theory supporting approaches for relative
> > > feature importance. However, if Shapley values for XAI were somehow to
> > > miraculously work for "functions used in the relevant literature",
> > > that should now be proved, given the now proved fact that Shapley
> > > values for XAI are generally unsound.

---

### Official Review · Reviewer_HAap · 2023-07-17

**Soundness:** 3 good
**Presentation:** 4 excellent
**Contribution:** 2 fair
**Rating:** 6
**Confidence:** 4

**Summary:**

Based on definitions of feature necessity, relevancy, and irrelevancy from previous work,as well as systematic issues with Shapley values for explainability on boolean classifiers (e.g. non-zero Shapley values assigned to irrelevant features, zero Shapley values assigned to relevant features, among others) identified in previous work, the authors offer proof for their existence in functions with an arbitrary number of variables. They conclude that the existence of such systematic issues is cause for concern in using Shapley values for explainability, as misleading information about feature importance can induce errors in human decision making.


**Strengths:**

- Originality: The work offers proof for the existence of issues with Shapley value explanations on boolean functions with an arbitrary number of variables that were previously only studied empirically.
- Quality and Clarity: The theoretical framework, preliminaries, and proofs are described in a very concise manner. Despite the theoretical nature of the paper, the authors are able to concisely state to the reader what is described in each formula (e.g. lines 125-127: Thus, given an instance (v, c), a (weak) AXp is a subset of features which, if fixed to the values dictated by v, then the prediction is guaranteed to be c, independently of the values assigned to the other features). Similarly, the main idea for each proof is described in a very intuitive manner, increasing readability of the paper significantly.
- Significance: The present work proves systematic issues exhibited by Shapley value explanations on boolean functions. Shapley values are one of the most popular solutions, as they are based on clearly defined axioms, i.e. properties deemed desirable for explanations. For boolean functions, the present work shows that these axioms (which Shapley values do fulfill) may be lacking for treating irrelevant and relevant features as would be expected.

**Weaknesses:**

I am a bit concerned with the novelty, as the present work only provides proof for observations about unexpected behavior of Shapley value explanations for boolean functions that were already observed empirically in previous work (however, the authors also state themselves that these issues have been identified empirically in previous work). To raise concern about e.g. I1, it would be sufficient to simply identify a case where irrelevant values are assigned nonzero Shapley values.

I also believe the title promises a bit more than is provided by the paper. The proofs and resulting claims are restricted to boolean functions, however, it would be interesting to see how and if the described issues occur in continuous settings, e.g., when explaining DNNs.

**Questions:**

I would be interested in how the findings translate to non-boolean functions, i.e., continuous setting. E.g. the definitions of relevant, irrelevant, and necessary features over classification change makes sense in the boolean setting, but not in a continuous setting where logit values easily change with features being added.

Also, it would be interesting to see some concrete recommendations as to how the identified issues might be avoided.

**Limitations:**

restriction to boolean functions, as described in "Weaknesses" section. I think a paragraph of how the described proofs and observations may impact Shapley value explanations in more real-world settings would go a long way here, as well as suggestions on how to mitigate the proven issues.

---

> ### Author Rebuttal · Authors · 2023-08-09
>
> We underscore that our paper extends significantly the experimental
> results from [35]. The proofs that there exist arbitrary many boolean
> classifiers for which Shapley values give misleading relative feature
> importance offer a strong argument for why these results should be
> presented to a wider audience. Also, we expand the issues reported
> in [35], and prove that these also exist for arbitrary many boolean
> classifiers. Moreover, and to the best of our knowledge, the earlier
> experimental work (i.e. reference [35]) has not been presented at any
> conference. Given the importance of a result that refutes the validity
> of Shapley values in explainability, we believe that such a result
> should be made widely visible by presentation in a top-tier
> conference, especially when the refutation is being established for
> arbitrary many (boolean) classifiers.
>
>
> Answer to questions:
>
> Q1: First, the empirical classifier search described in earlier work
> (i.e. reference [35]) can hardly serve as the basis to answer this
> question. Second, the techniques proposed in our paper can be adapted
> to prove results for non-boolean cases. This requires understanding
> the proof techniques that we propose, and then considering
> generalizations to non-boolean functions with categorical
> features. This is the subject of future work.
>
> Q2: By understanding the limitations of existing definitions of
> Shapley values, it is now possible to modify those definitions in
> order to address those limitations. For example, there is recent work
> proposing an alternative to Shapley values, that relates with the
> research detailed on our paper. This is described in the following
> preprint:
> J. Yu, A. Ignatiev, P. J. Stuckey: On Formal Feature Attribution and
> Its Approximation. CoRR abs/2307.03380 (2023)
>
> An open direction of research, not addressed in the preprint above, is
> how to adapt the definition of Shapley values such that (ir)relevancy
> of features is accounted for.

---

### Decision · Program_Chairs · 2023-09-21

**Decision:**

Reject

**Comment:**

Reviewers criticise the limited contribution (especially wrt previous work such as [35]), the unclear practical impact (e.g. explaining DNNs) and the conclusions drawn from the paper (e.g.,  Reviewer HAap "I also believe the title promises a bit more than is provided by the paper." or Reviewer 4AZv "It may be that Shapley values both ARE useful and legitimate for ML explainability AND do not track necessary or irrelevant features."). In my opinion this paper is a clear borderline paper. However, because of the strong criticism I rather tend towards rejecting it.